# Ground-based lidar processing and simulator framework for comparing models and observations (ALCF 1.0)

Peter Kuma[1], Adrian J. McDonald[1], Olaf Morgenstern[2], Richard Querel[3], Israel Silber[4], and Connor J. Flynn[5]

[1]University of Canterbury, Christchurch, New Zealand
[2]National Institute of Water & Atmospheric Research (NIWA), Wellington, New Zealand
[3]National Institute of Water & Atmospheric Research (NIWA), Lauder, New Zealand
[4]Department of Meteorology and Atmospheric Science, Pennsylvania State University, PA, USA
[5]School of Meteorology, University of Oklahoma, Norman, OK, USA

**Correspondence:** Peter Kuma (peter@peterkuma.net)

**Abstract.** Automatic lidars and ceilometers provide valuable information on cloud and aerosols, but have not been used systematically in the evaluation of GCMs and NWP models. Obstacles associated with the diversity of instruments, a lack of standardisation of data products and open processing tools mean that the value of the large ALC networks worldwide is not being realised. We discuss a tool, called the Automatic Lidar and Ceilometer Framework (ALCF), that overcomes these problems and also includes a ground-based lidar simulator, which calculates the radiative transfer of laser radiation, and allows one-to-one comparison with models. Our ground-based lidar simulator is based on the Cloud Feedback Model Intercomparison Project (CFMIP) Observation Simulator Package (COSP) which has been used extensively for spaceborne lidar intercomparisons. The ALCF implements all steps needed to transform and calibrate raw ALC data and create simulated attenuated volume backscattering coefficient profiles for one-to-one comparison and complete statistical analysis of cloud. The framework supports multiple common commercial ALCs (Vaisala CL31, CL51, Lufft CHM 15k and Droplet Measurement Technologies MiniMPL), reanalyses (JRA-55, ERA5 and MERRA-2) and models (AMPS and the Unified Model). To demonstrate its capabilities, we present case studies evaluating cloud in the supported reanalyses and models using CL31, CL51, CHM 15k and MiniMPL observations at three sites in New Zealand. We show that the reanalyses and models generally underestimate cloud fraction. If sufficiently high temporal resolution model output is available (better than 6 hourly), a direct comparison of individual clouds is also possible. We demonstrate that the ALCF can be used as a generic evaluation tool to examine cloud occurrence and cloud properties in reanalyses, NWP models and GCMs, potentially utilising the large amounts of ALC data already available. This tool is likely to be particularly useful for the analysis and improvement of low-level cloud simulations which are not well monitored from space. This has previously been identified as a critical deficiency in contemporary models, limiting the accuracy of weather forecasts and future climate projections. While the current focus of the framework is on clouds, support for aerosol in the lidar simulator is planned in the future.

# 1 Introduction

Automatic lidars and ceilometers (ALC) are active ground-based instruments which emit laser pulses in the ultraviolet, visible or infrared (IR) part of the electromagnetic spectrum and measure radiation backscattered from atmospheric constituents such as cloud and fog liquid droplets and ice crystals, haze, aerosol and atmospheric gases (Emeis, 2010). Vertical profiles of attenuated backscattered radiation can be produced by measuring received power as a function of time elapsed between emitting the pulse and receiving the backscattered radiation. Quantities such as cloud base height (CBH) and a cloud mask (Pal et al., 1992; Wang and Sassen, 2001; Martucci et al., 2010; Costa-Surós et al., 2013; Van Tricht et al., 2014; Liu et al., 2015a, b; Lewis et al., 2016; Cromwell and Flynn, 2018; Silber et al., 2018), particle volume backscattering coefficient (Marenco et al., 1997; Welton et al., 2000, 2002; Wiegner and Geiß, 2012; Wiegner et al., 2014; Jin et al., 2015; Dionisi et al., 2018) and boundary layer height (Eresmaa et al., 2006; Münkel et al., 2007; Emeis et al., 2009; Tsaknakis et al., 2011; Milroy et al., 2012; Knepp et al., 2017) can be derived from the attenuated volume backscattering coefficient profile. Lidars equipped with polarisation or multiple wavelengths can also provide depolarisation ratio or colour ratio, respectively, which can be used to infer cloud phase or particle types. Doppler lidars can measure wind speed in the direction of the lidar orientation. ALCs are commonly deployed at airports, where they provide CBH, fog and aerosol observations needed for air traffic control. Large networks of up to hundreds of lidars and ceilometers have been deployed worldwide: Cloudnet (Illingworth et al., 2007), E-PROFILE (Illingworth et al., 2018), PollyNET (Baars et al., 2016), ICENET (Cazorla et al., 2017), MPLNET (Welton et al., 2006) and ARM (Stokes and Schwartz, 1994; Campbell et al., 2002). The purpose of these networks is to observe cloud, fog, aerosol, air quality, visibility and volcanic ash, provide input to numerical weather prediction (NWP) model evaluation (Hogan et al., 2001; Illingworth et al., 2007; Morcrette et al., 2012; Warren et al., 2018; Lamer et al., 2018; Hansen et al., 2018b) and assimilation (Illingworth et al., 2015b, 2018) and for climate studies. These networks are usually composed of multiple types of ALCs, with Vaisala CL31, CL51, Lufft (formerly Jenoptik) CHM 15k and Droplet Measurement Technologies (formerly Sigma Space and Hexagon) MiniMPL the most common. Complex lidar data processing has been set up on some of these networks. Notably, at the SIRTA site in France, lidar ratio (LR) comparable with a lidar simulator (Chiriaco et al., 2018) is calculated as part of their "ReOBS" processing method. Intercomparison and calibration campaigns such as CeiLinEx2015 (Mattis et al., 2016) and INTERACT-I(-II) (Rosoldi et al., 2018; Madonna et al., 2018) have been performed. Lidar data processing involves a number of tasks such as resampling, calibration, noise removal and cloud detection. Some of these are implemented in the instrument firmware of ALCs. This, however, means that lidar attenuated volume backscattering coefficient and detected cloud and cloud base are not comparable between different instruments. In most cases the algorithms are not publicly documented, making it impossible to compare the data with values from a model or a lidar simulator without a systematic bias.

Atmospheric model evaluation is an ongoing task, and a critical part of the model improvement process (Eyring et al., 2019; Hourdin et al., 2017; Schmidt et al., 2017). Traditionally, various types of observational and model datasets have been utilised – weather and climate station data, upper air soundings, ground-based and satellite remote sensing datasets, and high-resolution model simulations, amongst others. Clouds are one of the most problematic phenomena in atmospheric models due to their transient nature, high spatial and temporal variability, and sensitivity to a complex combination of conditions such as relative

humidity, aerosols (presence of cloud condensation nuclei and ice nuclei), thermodynamic and dynamic conditions. At the same time, clouds have a very substantial effect on the atmospheric shortwave and longwave radiation balance, and any cloud misrepresentation has a strong effect on other components of the model, limiting the ability to accurately represent past and present climate and predict future climate (Zadra et al., 2018). An improved understanding of clouds and cloud feedbacks is one of the focuses of the Coupled Model Intercomparison Project Phase 6 (CMIP6) (Eyring et al., 2016), and comparison of model cloud with observations is one of the key points of the Cloud Feedback Model Intercomparison Project (CFMIP) (Webb et al., 2017). Satellite observations make up the majority of the data used to evaluate model clouds. These include passive visible and IR low earth orbit and geostationary radiometers measuring, among others, features such as cloud cover, cloud top height (CTH), cloud top temperature; passive microwave instruments measuring total column water; active radars and lidars measuring cloud vertical profiles. Ground-based remote sensing instruments include radars, lidars, ceilometers, radiometers and sky cameras. As pointed out by Williams and Bodas-Salcedo (2017), using a wide range of different observational datasets including satellite and ground-based for general circulation model (GCM) evaluation is important due to limitations of each dataset.

Model cloud is commonly represented by the mixing ratio of liquid and ice and cloud fraction (CF) on every model grid cell and vertical level. In addition some models provide the cloud droplet effective radius used in radiative transfer calculations. Remote sensing observations do not match the representation of the atmospheric model fields directly because of their different resolutions, limited field of view (FOV) and attenuation by atmospheric constituents before reaching the instrument's receiver. Instrument simulators bridge this gap by converting the model fields to quantities which emulate those measured by the instrument, which can then be compared directly with observations. One such collection of instrument simulators is the CFMIP Observation Simulator Package (COSP) (Bodas-Salcedo et al., 2011; Swales et al., 2018), which has been used for more than a decade for evaluation of models using satellite, and more recently ground-based, observations. The simulators in COSP include active instruments such as spaceborne and ground-based radars: Cloud Profiling Radar (CPR) on CloudSat (Stephens et al., 2002), Ka-band ARM Zenith Radar (KAZR); lidars: Cloud-Aerosol Lidar Orthogonal Polarization (CALIOP) on CALIPSO (Winker et al., 2009), Cloud-Aerosol Transport System (CATS) on ISS (McGill et al., 2015), the Atmospheric Lidar (ATLID) on EarthCARE (Illingworth et al., 2015a); and spaceborne passive instruments: ISCCP (Rossow and Schiffer, 1991), MODIS (Parkinson, 2003) and MISR (Diner et al., 1998). The more recent addition of ground-based radar (Zhang et al., 2018) and lidar (Chiriaco et al., 2018; Bastin et al., 2018) opens up new possibilities to use the large amount of remote sensing data obtained from ground-based active remote sensing instruments. In practice, ground-based observational remote sensing data are not straightforward to use without a substantial amount of additional processing. Some previous studies have also compared models and ground-based radar and lidar observations without the use of an instrument simulator (Bouniol et al., 2010; Hansen et al., 2018a), though for the reasons identified above this is not advisable.

In this study we introduce a software package called the Automatic Lidar and Ceilometer Framework (ALCF) for evaluating model cloud using ALC observations. It extends and integrates the COSP lidar simulator (Chiriaco et al., 2006; Chepfer et al., 2007, 2008) with pre- and post-processing steps, and allows the simulator to be run offline on model output, instead of having to be integrated inside the model. This makes it possible to compare ALC data at any location without having to run the model

with a specific configuration. Multiple ALCs, reanalyses and model output formats are supported. The original COSP lidar simulator was extended with Rayleigh, Mie and ice crystal scattering at multiple lidar wavelengths. Observational ALC data from a number of common instruments can be processed by re-sampling to a common resolution, removing noise, detecting cloud and calculating statistics. The same steps can be performed on the simulated lidar data from the model (the output of running COSP on the model data), allowing for one-to-one comparison of model and observations. A particular focus of our work was on applying the same processing steps on the observed and simulated attenuated volume backscattering coefficient in order to avoid biases. The ALCF is made available under an open source license (MIT) at https://alcf-lidar.github.io, and as a permanent archive of code and technical documentation on Zenodo at https://doi.org/10.5281/zenodo.4088217.

A relatively small amount of other open source code is available for ALC data processing. A lidar simulator has been developed as part of the Goddard Satellite Data Simulator Unit (G-SDSU) (G-S, 2019), a package based on the instrument simulator package SDSU (Masunaga et al., 2010). The Community Intercomparison Suite (CIS) (Watson-Parris et al., 2016) allows for subsetting, aggregation, co-location and plotting of mostly satellite data with a focus on model–observations intercomparison. The STRAT lidar data processing tools are a collection of tools for conversion of raw ALC data, visualisation and feature classification (Morille et al., 2007).

Here, we provide an overview of the ALCF (Sect. 2), describe the supported ALCs, reanalyses and models (Sect. 3), the lidar simulator (Sect. 4) and the observed and simulated lidar data processing steps (Sect. 5). Later, we present a set of case studies at three sites in New Zealand (NZ) (Sect. 6) to demonstrate the value of this new tool. Lastly, we present the results of the case studies in Sect. 7.

## 2 Overview of operation of the Automatic Lidar and Ceilometer Framework (ALCF 1.0)

The ALCF performs the necessary steps to simulate ALC attenuated volume backscattering coefficient based on 4-dimensional atmospheric fields from reanalyses, NWP models and GCMs, and to transform the observed raw ALC attenuated volume backscattering coefficient profiles to profiles comparable with the simulated profiles. It does so by extracting 2-dimensional (time × height) profiles from the model data, performing radiative transfer calculations based on a modified COSP lidar simulator (Sect. 4), absolute calibration and resampling of the observed attenuated volume backscattering coefficient to common resolution and performing comparable cloud detection on the simulated and observed attenuated volume backscattering coefficient. The framework supports multiple common ALCs (Sect. 3.1), reanalyses and models (Sect. 3.2). The schematic in Fig. 1 illustrates this process as well as the ALCF commands which perform the individual steps. The following commands are implemented: **model**, **simulate**, **lidar**, **stats** and **plot**. The commands are normally executed in a sequence, which is also implemented by a meta-command **auto**, which is equivalent to executing a sequence of commands. The commands are described in detail in the technical documentation available online at https://alcf-lidar.github.io, on Zenodo at https://doi.org/10.5281/zenodo.4088217 and in the Supplementary information. The physical basis is described here.

The **model** command extracts 2-dimensional profiles of cloud liquid and ice content (and other thermodynamic fields) from the supported NWP model, GCM and reanalysis data (*model data* in Fig. 1) at a geographical point, along a ship track or a

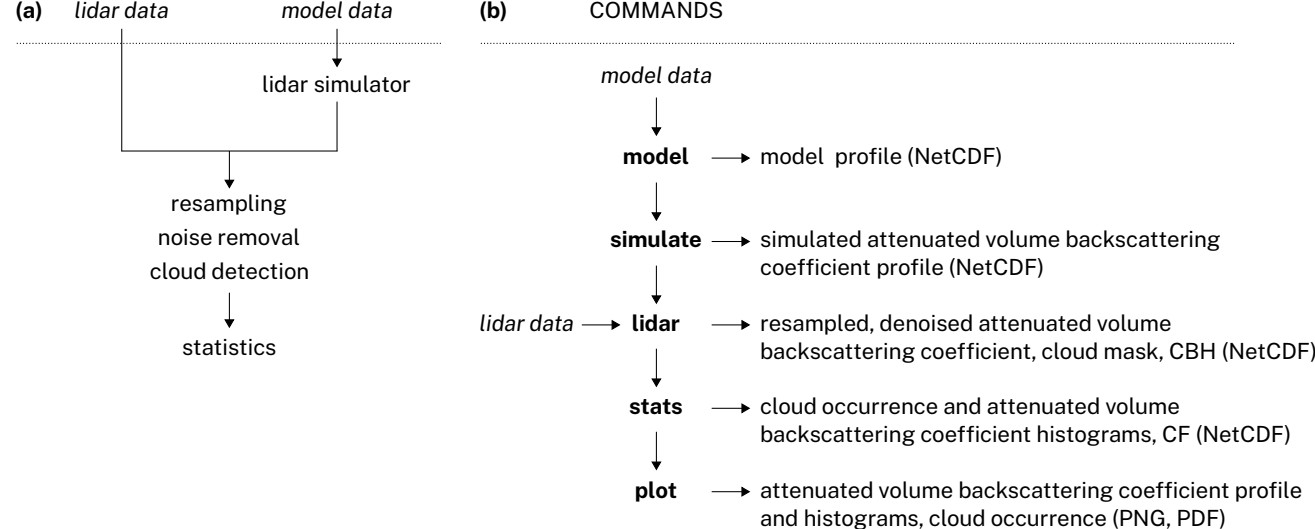

**Figure 1. (a)** Scheme showing the operation of the ALCF and **(b)** the processing commands.

flight path. The resulting profiles are recorded as NetCDF files. Sect. 3.2 describes the supported reanalyses and models. The *model data* can be either in one of the supported model output formats, or a new module for reading arbitrary model output can be written, providing that the required atmospheric fields are present in the model output. The required model fields are: per-level specific cloud liquid water content, specific cloud ice water content, cloud fraction, geopotential height, temperature,

surface-level pressure and orography. No physical calculations are performed by this command. The atmospheric profiles are extracted by a nearest-neighbour selection.

The **simulate** command runs the lidar simulator described in Sect. 4 on the extracted model data (the output of the **model** command) and produces simulated attenuated volume backscattering coefficient profiles. This command runs the COSP-derived lidar simulator, which performs radiative transfer calculations of the laser radiation through the atmosphere. The

resulting simulated attenuated volume backscattering coefficient profiles are the output of this command.

The **lidar** command applies various processing algorithms on either the simulated attenuated volume backscattering coefficient (the output of the **simulate** command) or the observed ALC coefficient (*lidar data* in Fig. 1) (Sect. 5). The data are resampled to increase the signal-to-noise ratio (SNR), noise is subtracted, LR is calculated, a cloud mask is calculated by applying a cloud detection algorithm and CBH is determined from the cloud mask. Absolute calibration (Sect. 5.2) can also be

applied in this step by multiplying the observed attenuated volume backscattering coefficient by a calibration coefficient. This is important in order to obtain unbiased attenuated volume backscattering coefficient profiles comparable with the simulated profiles. Sect. 3.1 describes the supported instruments. The *lidar data* can be in one of the supported instrument formats. If the native instrument format is not NetCDF, it has to be converted from the native format with the auxiliary command **convert** or one of the conversion programs: cl2nc (Vaisala CL31, CL51), mpl2nc or SigmaMPL (Sigma Space MiniMPL).

**Table 1.** Table of ALCs and their technical parameters. Power is calculated as Pulse $\times$ Pulse Repetition Frequency (PRF).

| Instrument | $\lambda$ (nm) | Laser | Rate[1] (s) | Res.[2] (m) | Depol.[3] | Pulse[4] ($\mu$J) | Range[5] (km) | PRF (kHz) | Overlap[6] (m) | Power (mW) | FOV[7] ($\mu$rad) |
|---|---|---|---|---|---|---|---|---|---|---|---|
| CHM 15k | 1064 | Nd:YAG | 2–600 | 5 | no | 7–9 | 15.4 | 5–7 | 1000[8] | 48 | 450 |
| CL31 | 910 | InGaAs | 2–120 | 10 | no | 1.2 | 7.7 | 10 | 70[8] | 12 | 830 |
| CL51 | 910 | InGaAs | 6–120 | 10 | no | 3 | 15.4 | 6.5 | 230[9] | 20 | 560 |
| MiniMPL | 532 | Nd:YAG | 1–900 | 5–75 | yes | 3–4 | 30.0 | 2.5 | 2000[9] | 9 | 110 |

[1] Sampling rate. [2] Vertical (range) resolution. [3] Depolarisation. [4] Pulse energy. [5] Maximum range. [6] Range of full overlap. [7] Receiver field of view. [8] Hopkin et al. (2019). [9] Madonna et al. (2018).

The **stats** step calculates summary statistics from the output of the **lidar** command. These include CF, cloud occurrence by height, attenuated volume backscattering coefficient histograms and the averages of LR and backscattering coefficient.

The **plot** command plots attenuated volume backscattering coefficient profiles produced by the **lidar** command (Fig. 4, 5, 6), and the statistics produced by the **stats** command: cloud occurrence (Fig. 3), attenuated volume backscattering coefficient histograms (Fig. 7) and attenuated volume backscattering coefficient noise standard deviation histograms (Fig. 9).

## 3 Supported input data: instruments, reanalyses and models

### 3.1 Instruments

The primary focus of the framework is to support common commercial ALCs. Ceilometers are considered the most basic type of lidar (Emeis, 2010; Kotthaus et al., 2016) intended as commercial products designed for unattended operation. They are used routinely to measure CBH, but most instruments also provide the full vertical profiles of attenuated volume backscattering coefficient. Therefore, they are suitable for model evaluation by comparing not only CBH, but also cloud occurrence as a function of height. Their compact size and low cost make it possible to deploy a large number of these instruments in different locations, or use them in unusual settings such as mounted on ships (Klekociuk et al., 2019; Kuma et al., 2020). Common off-the-shelf ceilometers are the Lufft CHM 15k, Vaisala CL31 and CL51. Some lidars offer higher power and therefore higher SNR, and capabilities not present in ceilometers such as dual polarisation, multiple wavelengths, Doppler shift measurement and Raman scattering. Below we describe ALCs supported by the framework and used in our case studies: Lufft CHM 15k, Vaisala CL31 and CL51 and Droplet Measurement Technologies MiniMPL. Table 1 lists selected parameters of the supported ALCs.

*Lufft CHM 15k* (previously Jenoptik CHM 15k) is a ceilometer operating at a wavelength of 1064 nm (near IR). The maximum range of the instrument is 15.4 km, vertical sampling resolution 5 m in the first 150 m and 15 m above and sampling rate 2 s. The total number of vertical levels is 1024. The wavelength in the near IR spectrum ensures low molecular backscattering. The instrument produces NetCDF files containing uncalibrated attenuated volume backscattering coefficient profiles and various derived variables, although the calibration coefficient is relatively consistent for different instruments of the model (Hopkin et al., 2019, Fig. 13).

**Table 2.** Reanalyses and models used in the case studies and some of their main properties. The temporal and horizontal grid resolution and vertical levels listed is the resolution of the model output available. The horizontal grid resolution is determined at 45°S. The internal resolution of the model may be different (see Sect. 3.2 for details). The reanalyses and the UM use regular longitude-latitude grids, while the AMPS horizontal grid is regular in the South Pole stereographic projection.

| Model/Grid | Type | Time resolution | Horizontal grid resolution | Vertical levels |
|---|---|---|---|---|
| AMPS/D01 | NWP | 3 h | $0.27° \times 0.19°$ (21×21 km) | 60 |
| ERA5 | Reanalysis | 1 h | $0.25° \times 0.25°$ (20×28 km) | 37 |
| JRA-55 | Reanalysis | 6 h | $1.25° \times 1.25°$ (98×139 km) | 37 |
| MERRA-2 | Reanalysis | 3 h | $0.625° \times 0.50°$ (49×56 km) | 72 |
| UM (GA7.1)/N96 | GCM | 20 min. | $1.875° \times 1.25°$ (147×139 km) | 85 |

*Vaisala CL31 and CL51* are ceilometers operating at a wavelength of 910 nm (near IR). The maximum range of CL31 and CL51 is 7.7 km and 15.4 km and the sampling rate is 2 and 6 s, respectively. The vertical resolution is 10 m. The total number of vertical levels is 770 and 1540, respectively. The wavelength is characterised by relatively low molecular backscattering (but higher than 1064 nm) and is affected by water vapour absorption (Wiegner and Gasteiger, 2015; Wiegner et al., 2019), which

can cause additional absorption of about 20% in the mid-latitudes and 50% in the tropics (see also Sect. 5.4). The instruments produce data files containing uncalibrated attenuated volume backscattering coefficient which can be converted to NetCDF (see cl2nc in the Code and data availability section). The firmware configuration option "noise_h2 off" results in backscatter range correction to be selectively applied under a certain critical range and above this range only if cloud is present (Kotthaus et al., 2016, Sect. 3.2). This was the case with our case study dataset (Sect. 6). We apply range correction on the uncorrected

range gates during lidar data processing. The critical range in CL51 is not documented, but was determined as 6000 m based on an observed discontinuity.

*Droplet Measurement Technologies Mini Micro Pulse Lidar (MiniMPL)* (previously Sigma Space MiniMPL and Hexagon MiniMPL) (Spinhirne, 1993; Campbell et al., 2002; Flynn et al., 2007) is a dual-polarisation micro pulse lidar (meaning that it uses a high pulse repetition rate (PRF) and low pulse power) operating at a wavelength of 532 nm (green colour in the visible

spectrum). The maximum range of the instrument is 30 km. The vertical resolution is 5–75 m and sampling rate 1 s. The shorter wavelength is affected by stronger molecular backscattering than 910 nm and 1064 nm. The instrument can be housed in an enclosure with a scanning head to provide configurable scanning by elevation angle and azimuth. The instrument produces data files containing raw attenuated volume backscattering coefficient which can be converted to NetCDF containing normalised relative backscatter (NRB) with a vendor-provided tool SigmaMPL (see also mpl2nc in the Code and data availability section).

## 3.2 Reanalyses and models

Below we briefly describe reanalyses and models[1] used in the case studies presented here (Sect. 6). We used publicly available output from three reanalyses and one NWP model. In addition, we performed nudged GCM simulations with high-temporal resolution output with the Unified Model (UM). Table 2 lists some of the main properties of the reanalyses and models.

*The Antarctic Mesoscale Prediction System (AMPS)* (Powers et al., 2003) is a limited-area NWP model based on the polar fifth-generation Pennsylvania State University-National Center for Atmospheric Research Mesoscale Model (Polar MM5), now known as the Polar Weather Research and Forecasting (WRF) model (Hines and Bromwich, 2008). The model serves operational and scientific needs in Antarctica, but its largest grid also covers the South Island of NZ. AMPS forecasts are publicly available on the Earth System Grid (Williams et al., 2009). The forecasts are produced on several domains. The largest domain D01 used in the presented analysis covers NZ and has horizontal grid spacing of approximately 21 km over NZ. The model uses 60 vertical levels. The model output is available in 3-hourly intervals and initialised at 00:00 and 12:00 UTC. The initial and boundary conditions are based on the Global Forecasting System (GFS) global NWP model. AMPS assimilates local Antarctic observations from human-operated stations, automatic weather stations (AWS), upper-air stations and satellites.

*ERA5* (ECMWF, 2019) is a reanalysis produced by the European Centre For Medium-Range Weather Forecasts (ECMWF) currently available for the time period 1979 to present, with a plan to extend the time period to 1950. The reanalysis is based on the global NWP model Integrated Forecast System (IFS) version CY41R2. It uses a 4D-Var assimilation of station, satellite, radiosonde, radar, aircraft, ship-based and buoy data. The model has 137 vertical levels. Atmospheric fields are interpolated from horizontal resolution equivalent of 31 km and 137 model levels on regular longitude-latitude grid of 0.25° and 37 pressure levels, and made available to the end-users. In this analysis we use the hourly data on pressure and surface levels.

*Japanese 55-year reanalysis (JRA-55)* (Ebita et al., 2011; Kobayashi et al., 2015; Harada et al., 2016) is a global reanalysis produced by the Japan Meteorological Agency (JMA) and the Central Research Institute of Electric Power Industry (CRIEPI) based on the JMA Global Spectral Model (GSM). The reanalysis is available from 1958 onward. The reanalysis is based on the JMA operational assimilation system. JRA-55 uses a 4D-Var assimilation of surface, upper-air, satellite, ship-based and aircraft observations. The model uses 60 vertical levels and a horizontal grid with resolution approximately 60 km. In this analysis we use the 1.25° isobaric analysis and forecast fields interpolated to 37 pressure levels.

*Modern-Era Retrospective analysis for Research and Applications (MERRA-2)* (Gelaro et al., 2017) is a reanalysis produced by the NASA Global Modeling and Assimilation Office (GMAO). The reanalysis is based on the Goddard Earth Observing System (GEOS) atmospheric model. The model has approximately 0.5°×0.65° horizontal resolution and 72 vertical levels. It performs 3D-Var assimilation of station, upper-air, satellite, ship-based and aircraft data in 6-hourly cycles. In this analysis, we use the MERRA-2 3-hourly instantaneous model-level assimilated meteorological fields (M2I3NVASM) version 5.12.4 product.

---

[1]We use the term "reanalysis" when referring to ERA5, JRA-55 and MERRA-2 even though the reanalyses are based on atmospheric models. We use the term "model" when referring to AMPS and the UM, which are atmospheric models.

*The UK Met Office Unified Model (UM)* (Walters et al., 2019) is an atmospheric model for weather forecasting and climate projection developed by the UK Met Office and the Unified Model Partnership. The UM is the atmospheric component, called Global Atmosphere (GA), of the HadGEM3–GC3.1 GCM and the UKESM1 earth system model (ESM). In this analysis we performed custom nudged runs of the UM (Telford et al., 2008) in the GA7.1 configuration with 20 min. time step and output temporal resolution on a New Zealand eScience Infrastructure (NeSI)/National Institute of Water & Atmospheric Research (NIWA) supercomputer (Williams et al., 2016). The model was nudged to the ERA-Interim (Dee et al., 2011) atmospheric fields of horizontal wind speed and potential temperature and the HadISST sea surface temperature (SST) and sea ice dataset (Rayner et al., 2003). The model uses 85 vertical levels and a horizontal grid resolution of $1.875° \times 1.25°$.

## 4   Lidar simulator

The COSP lidar simulator Active Remote Sensing Simulator (ACTSIM) was introduced by Chiriaco et al. (2006) for the purpose of deriving simulated CALIOP measurements (Chepfer et al., 2007, 2008). The simulation is implemented by applying the lidar equation on model levels. Scattering and absorption by cloud particles and air molecules is calculated using the Mie and Rayleigh theory, respectively. Scattering and absorption by aerosols is not implemented in the presented version, but support is planned in the future for models which provide concentration of aerosols. Therefore, the current focus of the simulator is solely on cloud evaluation. CALIOP operates at a wavelength of 532 nm, and calculations in the original COSP simulator use this wavelength. We implemented a small set of changes to the lidar simulator to support a number of ALCs with different operating wavelengths and developed parametrisation of backscattering from ice crystals based on temperature.

The lidar equation (Emeis, 2010) is based on the radiative transfer equation (Goody and Yung, 1995; Liou, 2002; Petty, 2006; Zdunkowski et al., 2007), which relates transmission of radiation to scattering, emission and absorption in media such as the atmosphere. The lidar equation assumes laser radiation passes through the atmosphere where it is absorbed and scattered. A fraction of laser radiation is scattered back to the instrument and reaches the receiver. Scattering and absorption in the atmosphere is determined by its constituents – gases, liquid droplets, ice crystals and aerosol particles. The focus of the current version of the simulator is on clouds. For this purpose, the atmospheric model output needed is 4-dimensional fields of mass mixing ratios of liquid and ice and CF. The lidar equation can be applied on these output fields to simulate the backscattered radiation received by the instrument. Table 3 lists physical quantities used in the following sections. Here, we use radiative transfer notation similar to Petty (2006) and the notation of the original lidar simulator (Chiriaco et al., 2006).

Below we provide a brief review of LR, Rayleigh and Mie scattering, calculate LR of cloud droplets at lidar wavelengths of the presented instruments and introduce an empirical parametrisation of LR and multiple scattering coefficient of ice crystals based on previous studies.

### 4.1   Lidar ratio

Lidar ratio $S$ is the extinction-to-backscattering ratio of atmospheric constituents at the lidar wavelength. It is an important quantity in lidar observations and the lidar simulator because it determines the amount of attenuation and backscattering.

**Table 3.** Table of physical quantities.

| Symbol | Name | Units | Expression |
|---|---|---|---|
| $\Omega$ | Solid angle | sr | |
| $z$ | Height relative to the instrument | m | |
| $k_{\mathrm{B}}$ | Boltzmann constant | $\mathrm{JK}^{-1}$ | $k_{\mathrm{B}} \approx 1.38 \times 10^{-23}\ \mathrm{JK}^{-1}$ |
| $p$ | Atmospheric pressure | Pa | |
| $T$ | Atmospheric temperature | K | |
| $\rho_{\mathrm{air}}$ | Air density | $\mathrm{kg.m}^{-3}$ | |
| $\rho$ | Liquid (or ice) density | $\mathrm{kg.m}^{-3}$ | |
| $q$ | Cloud liquid (or ice) mass mixing ratio | 1 | |
| $N$ | Particle number concentration | $\mathrm{m}^{-3}$ | |
| $\alpha_s$ ($\alpha_e$) | Volume scattering (extinction) coefficient | $\mathrm{m}^{-1}$ | |
| $P_\pi(\theta)$ | Scattering phase function at angle $\theta$ | 1 | $\int_{4\pi} P_\pi(\theta)\mathrm{d}\Omega = 4\pi$ |
| $\beta$ | Volume backscattering coefficient | $\mathrm{m}^{-1}\mathrm{sr}^{-1}$ | $\beta = \alpha_s P_\pi(\pi)/(4\pi)$ |
| $\beta_{\mathrm{mol}}$ | Volume backscattering coefficient for air molecules | | |
| $\beta_{\mathrm{p}}$ | Volume backscattering coefficient for cloud particles | | |
| $\eta$ | Multiple scattering coefficient | 1 | |
| $\beta'$ | Attenuated volume backscattering coefficient | $\mathrm{m}^{-1}\mathrm{sr}^{-1}$ | $\beta' = \beta \exp(-2\int_0^z \eta\alpha_e \mathrm{d}z)$ |
| $S$ | Lidar ratio (extinction-to-backscatter ratio) | sr | $S = \alpha_e/\beta$ |
| $S'$ | Effective (apparent) lidar ratio | sr | $S' = S\eta$ |
| $k$ | Backscatter-to-extinction ratio | $\mathrm{sr}^{-1}$ | $k = 1/S$ |
| $n(r)$ | Number distribution of particle size | $\mathrm{m}^{-4}$ | $N = \int_0^\infty n(r)\mathrm{d}r$ |
| $Q_s$ ($Q_e$) | Scattering (extinction) efficiency of spherical particles | 1 | $\alpha_s = Q_s\pi r^2 N,\ \alpha_e = Q_e\pi r^2 N$ |
| $Q_b$ | Backscattering efficiency of spherical particles | $\mathrm{sr}^{-1}$ | $\beta = Q_b\pi r^2 N$ |
| $r_{\mathrm{eff}}$ | Effective radius | m | $r_{\mathrm{eff}} = \int_0^\infty r^3 n(r)\mathrm{d}r / \int_0^\infty r^2 n(r)\mathrm{d}r$ |
| $\sigma_{\mathrm{eff}}$ | Effective standard deviation | m | $\sigma_{\mathrm{eff}} = \left(\int_0^\infty (r - r_{\mathrm{eff}})^2 r^2 n(r)\mathrm{d}r\right) / \left(\int_0^\infty r^2 n(r)\mathrm{d}r\right)$ |

LR is not explicitly known from the observed attenuated volume backscattering coefficient. For liquid cloud droplets at near IR wavelengths it is relatively constant at $S \approx 19$ sr (Sect. 4.2), while for ice crystals (Sect. 4.3) and aerosol it is highly variable. When the lidar signal is fully attenuated, and under the assumption that cloud LR is constant and scattering from clouds is much stronger than molecular and aerosol scattering, LR can be determined from the observed attenuated volume backscattering coefficient by integrating it vertically (O'Connor et al., 2004):

$$S' = \eta S = \frac{1}{2\int_0^\infty \beta' \mathrm{d}z}, \tag{1}$$

where $S'$ is effective (apparent) LR, a quantity which does not depend on the multiple scattering coefficient.

## 4.2 Rayleigh and Mie scattering

The Rayleigh volume backscattering coefficient $\beta_{\mathrm{mol}}$ ($\mathrm{m^{-1}sr^{-1}}$) in ACTSIM is parametrised by the following equation (Eq. (8) in Chiriaco et al. (2006)):

$$\beta_{\mathrm{mol}} = \frac{p}{k_{\mathrm{B}}T}(5.45 \times 10^{-32})\left(\frac{\lambda}{550\mathrm{nm}}\right)^{-4.09} = \frac{p}{k_{\mathrm{B}}T}C_{\mathrm{mol}}, \tag{2}$$

where for lidar wavelength $\lambda$ = 532 nm, $C_{\mathrm{mol}} = 6.2446 \times 10^{-32}$ ; $k_{\mathrm{B}}$ is the Boltzmann constant $k_{\mathrm{B}} \approx 1.38 \times 10^{-23}$ $\mathrm{JK^{-1}}$, $p$ is the atmospheric pressure and $T$ is the atmospheric temperature. We multiply this equation by $\exp(4.09(\log(532) - \log(\lambda)))$ (where the value of $\lambda$ is in nm) to get molecular backscattering for wavelengths other than 532 nm, which allows us to support multiple commercially available instruments. The strength of molecular backscattering is usually lower than backscattering from clouds for the relevant wavelengths.

The lidar signal at visible or near IR wavelengths is scattered by cloud droplets in the Mie scattering regime (Mie, 1908). In the most simple approximation, one can assume spherical dielectric particles. The scattering from these particles depends on the relative size of the wavelength and the (spherical) particle radius $r$, expressed by the dimensionless size parameter $x$:

$$x = \frac{2\pi r}{\lambda}. \tag{3}$$

While the wavelength is approximately constant during the operation of the lidar[2], the particle size comes from a distribution of sizes, typically approximated in NWP models and GCMs by a Gamma or log-normal distribution with a given mean and standard deviation. Some models provide the mean as effective radius $r_{\mathrm{eff}}$. If the effective radius is not provided by the model, the lidar simulator assumes a value $r_{\mathrm{eff}}$ = 10 µm by default, which is approximately consistent with global studies of effective radius (Bréon and Colzy, 2000; Bréon and Doutriaux-Boucher, 2005; Hu et al., 2007; Zhang and Platnick, 2011; Rausch et al., 2017; Fu et al., 2019). This is different from the default effective radius of 30 µm in the original COSP lidar simulator.

In order to support multiple laser wavelengths, it is necessary to calculate backscattering efficiency due to scattering by a distribution of particle sizes. We use the computer code MIEV developed by Warren J Wiscombe (Wiscombe, 1979, 1980) to calculate backscattering efficiency for a range of the size parameter $x$ and integrate for a distribution of particle sizes. The resulting pre-calculated LR (extinction-to-backscatter ratio) as a function of the effective radius is included in the lidar simulator for fast lookup during the simulation.

Cloud droplet size distribution parameters are an important assumption in lidar simulation due to the dependence of Mie scattering on the ratio of wavelength and particle size (the size parameter $x$). NWP models and GCMs traditionally use the effective radius $r_{\mathrm{eff}}$ and effective standard deviation $\sigma_{\mathrm{eff}}$ (or an equivalent parameter such as effective variance $\nu_{\mathrm{eff}}$) to parametrise this distribution. Knowledge of the real distribution is likely highly uncertain due to a large variety of clouds occurring globally

---

[2]The actual lidar wavelength is not constant and is characterised by a central wavelength and width. The central wavelength may fluctuate with temperature (Wiegner and Gasteiger, 2015).

**Table 4.** Table of sensitivity tests of theoretical distribution assumption, effective radius $r_{\text{eff}}$ and effective standard deviation $\sigma_{\text{eff}}$ of the cloud droplet size distribution. $\mu$ and $\sigma$ are the mean and standard deviation of a normal distribution corresponding to the log-normal distribution, calculated numerically from $r_{\text{eff}}$ and $\sigma_{\text{eff}}$. $\mu_*$ and $\sigma_*$ are the actual mean and standard deviation of the distribution (calculated numerically).

| Distribution | $r_{\text{eff}}$ ($\mu$m) | $\sigma_{\text{eff}}$ ($\mu$m) | $\mu$ | $\sigma$ | $\mu_*$ ($\mu$m) | $\sigma_*$ ($\mu$m) |
|---|---|---|---|---|---|---|
| log-normal | 20 | 10 | 2.44 | 0.47 | 12.76 | 6.26 |
| log-normal | 20 | 5 | 2.84 | 0.25 | 17.72 | 4.43 |
| log-normal | 10 | 5 | 1.74 | 0.47 | 6.40 | 3.20 |
| Gamma | 20 | 10 | | | 9.98 | 7.00 |
| Gamma | 20 | 5 | | | 17.50 | 4.68 |
| Gamma | 10 | 5 | | | 5.00 | 3.54 |

and the limited ability to predict microphysical cloud properties in models. In this section we introduce theoretical assumptions used in the lidar simulator based on established definitions of the effective radius and effective standard deviation and two common distributions. Edwards and Slingo (1996) discuss the effective radius in the context of model radiation schemes, and we will primarily follow the definitions detailed in Chang and Li (2001) and Petty and Huang (2011). The practical result of this section (and the corresponding offline code) is pre-calculated backscatter-to-extinction ratios as a function of the effective radius in the form of a lookup table included in the lidar simulator, and used in the online calculations. The offline code is provided and can be re-used for calculation of the necessary lookup tables for different lidar wavelengths, should the user of the code want to support another instrument.

The effective radius $r_{\text{eff}}$ and effective standard deviation $\sigma_{\text{eff}}$ are defined by:

$$r_{\text{eff}} = \frac{\int_0^\infty r^3 n(r)\,\mathrm{d}r}{\int_0^\infty r^2 n(r)\,\mathrm{d}r}, \quad \sigma_{\text{eff}}^2 = \frac{\int_0^\infty (r - r_{\text{eff}})^2 r^2 n(r)\,\mathrm{d}r}{\int_0^\infty r^2 n(r)\,\mathrm{d}r}, \tag{4}$$

where $n(r)$ is the probability density function (PDF) of the distribution. Here, we follow Petty and Huang (2011), who define the effective variance $\nu_{\text{eff}}$ which relates to $\sigma_{\text{eff}}$ by $\nu_{\text{eff}} = \sigma_{\text{eff}}^2 / r_{\text{eff}}^2$. Due to lack of knowledge about the real distribution of particle radii, it has to be modelled by a theoretical distribution, such as a log-normal or Gamma distribution. The original ACTSIM simulator assumes a log-normal distribution (Chiriaco et al., 2006) with the PDF:

$$n(r) \propto \frac{1}{r} \exp\left(-\frac{(\log r - \mu)^2}{2\sigma^2}\right), \tag{5}$$

where $\mu$ and $\sigma$ are the mean and the standard deviation of the corresponding normal distribution, respectively. Chiriaco et al. (2006) use the value of $\sigma = \log(1.2) = 0.18$ "for ice clouds" (the value for liquid cloud does not appear to be documented). In our parametrisation we used a combination of $r_{\text{eff}}$ and $\sigma_{\text{eff}}$ to constrain the theoretical distribution, where the effective standard deviation $\sigma_{\text{eff}}$ was assumed to be one fourth of the effective radius $r_{\text{eff}}$. This choice is approximately consistent with

$\sigma = \log(1.2) = 0.18$ at $r_{\text{eff}}$ = 20 $\mu$m (see Table 4, described below). In future updates, the values could be based on in situ studies of size distribution or taken from the atmospheric model output if available.

From the expression for the n-th moment of the log-normal distribution $E[X^n] = \exp(n\mu + n^2\frac{\sigma^2}{2})$ and Equation (4) we calculate $r_{\text{eff}}$ and $\sigma_{\text{eff}}$ of the log-normal distribution:

$$5 \quad r_{\text{eff}} = \frac{E[r^3]}{E[r^2]} = \exp(\mu + \frac{5}{2}\sigma^2), \tag{6}$$

$$\sigma_{\text{eff}}^2 = \frac{E[(r - r_{\text{eff}})^2 r^2]}{E[r^2]} = \frac{E[r^4] - 2E[r^3]r_{\text{eff}} + r_{\text{eff}}^2 E[r^2]}{E[r^2]} = \frac{\exp(4\mu + 8\sigma^2) - \exp(4\mu + 7\sigma^2)}{\exp(2\mu + 2\sigma^2)} =$$

$$= \exp(2\mu + 6\sigma^2) - \exp(2\mu + 5\sigma^2). \tag{7}$$

We find $\mu$ and $\sigma$ for given $r_{\text{eff}}$ and $\sigma_{\text{eff}}$ numerically by root-finding using the equations above. In practice, we find that the root-finding converges well for $r_{\text{eff}}$ between 5 and 50 $\mu$m, which is the range most likely to be applicable in practice.

The Gamma distribution follows the PDF:

$$n(r) \propto r^{(1-3\nu_{\text{eff}})/\nu_{\text{eff}}} \exp\left(-\frac{r}{r_{\text{eff}}\nu_{\text{eff}}}\right) \tag{8}$$

(see e.g. Eq. 13 in Petty and Huang (2011) or Eq. 1 in Bréon and Doutriaux-Boucher (2005)). In this case, the distribution depends explicitly on $r_{\text{eff}}$ and $\sigma_{\text{eff}}$, and as such does not require numerical root-finding.

Figure 2a shows the log-normal and Gamma distributions calculated for a number of $r_{\text{eff}}$ and $\sigma_{\text{eff}}$ values, and Table 4

summarises properties of these distributions. The actual mean and standard deviation of the distributions do not necessarily correspond well with the effective radius and effective standard deviation.

In ACTSIM, the volume extinction coefficient $\alpha_e$ is calculated by integrating the extinction by individual particles over the particle size distribution:

$$\alpha_e = \int_0^\infty Q_e \pi r^2 n(r) \mathrm{d}r \approx Q_e \pi \int_0^\infty r^2 n(r) \mathrm{d}r = Q_e \frac{3q\rho_{\text{air}}}{4\rho r_{\text{eff}}}, \tag{9}$$

assuming approximately constant extinction efficiency $Q_e \approx 2$ (which is approximately true for the interesting range of $r_{\text{eff}}$ and laser wavelengths), and using the relationship between the cloud liquid mass mixing ratio $q$ and $\int_0^\infty r^2 n(r) \mathrm{d}r$:

$$q\rho_{\text{air}} = \int_0^\infty \frac{4}{3}\pi r^3 \rho n(r) \mathrm{d}r = \frac{4}{3}\pi\rho \int_0^\infty r^3 n(r) \mathrm{d}r = \frac{4}{3}\pi\rho r_{\text{eff}} \int_0^\infty r^2 n(r) \mathrm{d}r, \tag{10}$$

where $\rho$ and $\rho_{\text{air}}$ are the densities of liquid water and air, respectively.

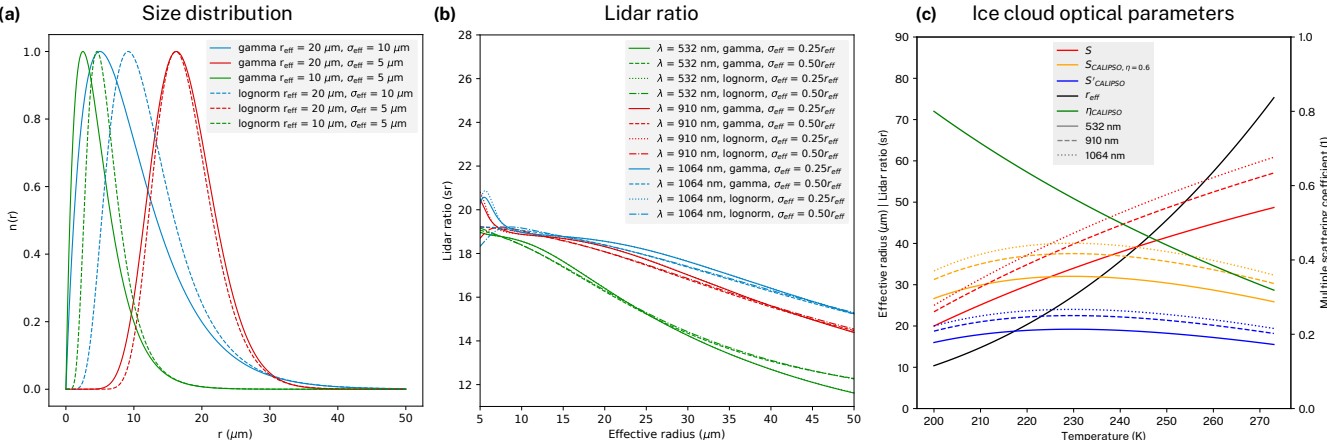

**Figure 2. (a)** Theoretical distributions of cloud droplet radius based on the log-normal and Gamma distributions parametrised by multiple choices of the effective radius $r_{\mathrm{eff}}$ and effective standard deviation $\sigma_{\mathrm{eff}}$. **(b)** Lidar ratio (LR) as a function of effective radius calculated for different theoretical cloud droplet size distributions, laser wavelengths and effective standard deviation ratios. **(c)** Parametrisation of ice cloud optical properties as a function of temperature based on Garnier et al. (2015) and Heymsfield (2005). The plot shows LR ($S$), LR of CALIPSO calculated using the constant standard processing multiple scattering coefficient $\eta = 0.6$ ($S_{\mathrm{CALIPSO},\eta=0.6}$), the effective LR of CALIPSO ($S'_{\mathrm{CALIPSO}}$), the effective radius ($r_{\mathrm{eff}}$), the multiple scattering coefficient of CALIPSO ($\eta_{\mathrm{CALIPSO}}$) determined by Garnier et al. (2015). LRs are calculated for three wavelengths of 532 nm (solid line), 910 nm (dashed line) and 1064 nm (dotted line) by scaling with colour ratio.

Likewise, the volume backscattering coefficient from particles $\beta_{\mathrm{p}}$ is calculated by integrating backscattering by individual particles over the particle size distribution:

$$\beta_{\mathrm{p}} = \int\limits_{0}^{\infty} Q_s \pi r^2 \frac{P_\pi(\pi)}{4\pi} n(r)\mathrm{d}r. \tag{11}$$

where $Q_s$ is scattering efficiency and $P_\pi(\pi)$ is scattering phase function at $180°$. Since the normalisation of $n(r)$ is not
5   known until the online phase of calculation, the backscatter-to-extinction ratio from particles $k_p = \beta/\alpha_e$ can be calculated offline instead (the requirement for normalisation of $n(r)$ is avoided by appearing in both the numerator and denominator):

$$k_p = \beta_{\mathrm{p}}/\alpha_e = \frac{\int_0^\infty Q_s r^2 P_\pi(\pi)/(4\pi)n(r)\mathrm{d}r}{\int_0^\infty Q_e r^2 n(r)\mathrm{d}r}. \tag{12}$$

We pre-calculate this integral numerically for a permissible interval of $r_{\mathrm{eff}}$ (5–50 µm) at 500 evenly spaced wavelengths, and store the result as a lookup table for the online phase. The integral in the numerator is numerically hard to calculate due to strong dependency of $P_\pi(\pi)$ on $r$. Figure 2b shows LR as a function of $r_{\mathrm{eff}}$, calculated for log-normal and Gamma particle size
10   distributions with $\sigma_{\mathrm{eff}} = 0.25 r_{\mathrm{eff}}$ and $\sigma_{\mathrm{eff}} = 0.5 r_{\mathrm{eff}}$. This corresponds to the lookup table we use in the online phase of the lidar

simulator. As can be seen in Fig. 2, LR depends only weakly on the choice of the distribution type and the effective standard deviation ratio.

## 4.3 Backscattering from ice crystals

Simulation of backscattering from ice crystals is relatively complex compared to backscattering from liquid droplets due to
the very high variability of ice crystal microphysical properties such as habit, size, orientation and surface roughness, all of
which affect LR, extinction cross section, single-scattering albedo and multiple scattering coefficient. Common habits include
hexagonal plates, hexagonal columns, hollow hexagonal columns, droxtals, bullet rosettes, hollow bullet rosettes and aggregates
(Baran, 2009; van Diedenhoven, 2017). Size can be highly variable and bimodal with a dependence on temperature and relative
humidity. Orientation is commonly random or horizontally oriented (often reported with hexagonal ice plates). Surface can vary
between smooth and rough depending on supersaturation and crystal age. In general, the Mie theory cannot be used to simulate
backscattering from ice crystals because of their irregular shape (Yang et al., 2014). While large crystals allow the use of the
geometric optics approximation to estimate the optical properties, smaller crystals and diffraction by large crystals necessitate
the use of more advanced techniques such as the T-matrix method, finite-difference time domain (FDTD), discrete dipole
approximation (DDA) and others, which are generally computationally expensive. Current global atmospheric models do not
normally parametrise the microphysical properties of cloud ice explicitly, and provide only very limited information such as
ice mass concentration and in some cases the effective radius of ice crystals in the model output. Radiative transfer schemes
of atmospheric models do not explicitly evaluate backscattering (the phase function at $180°$) and therefore cannot provide this
information to the simulator. Instead the phase function is parametrised by the asymmetry factor, which is likely insufficient to
give an accurate estimate of backscattering.

Because the model ice crystal microphysical and optical properties are not known, they have to be parametrised. A first
option is to parametrise the microphysical properties such as habit and size and calculate optical properties theoretically. A
second option is to parametrise the optical properties directly. This appears to be a more practical choice because of the broad
availability of global remote sensing measurements of optical properties from satellites and ground-based lidars, compared to
relatively scarce in situ measurements of ice crystals. Garnier et al. (2015) analysed CALIPSO lidar and co-located passive
infrared data from the Imaging Infrared Radiometer (IIR) and determined a global relationship between temperature and LR
and multiple scattering coefficient at the lidar wavelength of 532 nm. The multiple scattering coefficient is taken as a constant
of 0.6 in the standard CALIPSO data processing, but they identified that it is in fact variable between about 0.4 and 0.8. Here,
we parametrise LR based on their findings. LR varies with the lidar wavelength, a larger part of which is due to the change in
the diffraction peak and a smaller part is due to the variation of the refractive index (Borovoi et al., 2014). We use colour ratio
to estimate LR at lidar wavelengths other than 532 nm. Colour ratio of 1064 nm relative to 532 nm is commonly estimated for
the dual-wavelength lidars such as CALIOP. Here, we use a value of 0.8 approximately consistent with the results of Bi et al.
(2009) and Vaughan et al. (2010). The effective radius is defined for non-spherical particles as $r_{\mathrm{eff}} = \frac{3}{2}\frac{\mathrm{IWC}}{\sigma}$, where IWC is the
ice water content, and $\sigma$ is the volume extinction coefficient of ice. Heymsfield (2005) summarised ice crystal effective radius
(related to "IWC/$\sigma$" by a factor of 1.64) parametrised as a function of temperature based on a number of field studies. We use

this relationship for determination of the effective radius. Figure 2c shows the true and effective LR based on Garnier et al. (2015) and the effective radius based on Heymsfield (2005), parametrised by the following equations:

$$S = \left(20 + (34 - 20)\frac{1/T - 1/200}{1/230 - 1/200}\right) \text{sr},$$

$$\eta = 0.8 + (0.5 - 0.8)\frac{1/T - 1/200}{1/240 - 1/200},$$

$$r_{\text{eff}} = \exp\left(\log(16.4) + (\log(49.2) - \log(16.4))\frac{1/T - 1/213.15}{1/253.15 - 1/213.15}\right) \mu\text{m},$$

where $T$ is atmospheric temperature in K. $S$ follows (Garnier et al., 2015, Figure 12b), $\eta$ follows (Garnier et al., 2015, Figure 9a) and $r_{\text{eff}}$ follows (Heymsfield, 2005, Figure 2), where the concave and convex shape (respectively) as approximated by using $1/T$ as an argument of the linear approximation, and we use a logarithmic scale of $r_{\text{eff}}$ in the expression for $r_{\text{eff}}$ to avoid negative values at low temperature. Figure 2c also shows LR when calculated with the assumption of $\eta = 0.6$ ($S_{\text{CALIPSO},\eta=0.6}$) as in the standard processing of CALIPSO data. This corresponds to the empirically found relationship in (Garnier et al., 2015, Figure 12a) and (Josset et al., 2012, Figure 9) with a local maximum at 225 K. LR at wavelengths other than 532 nm is approximated by $0.8^{\frac{\lambda - 532}{532}}$, where $\lambda$ is lidar wavelength in $\mu$m and 0.8 is the approximate value of 1064 nm/532 nm colour ratio. The parametrisation of LR ($S$ in Fig. 2c) spans about the same range of values as reported by (Hopkin, 2018, Figure 5.6) (20 to 60 sr) and Yorks et al. (2011) (10 to 60 sr). Based on CALIPSO observations, Hu (2007) identified that while effective LR of global ice clouds at a lidar wavelength of 532 nm is mostly clustered around 17 sr, horizontally oriented plates produce a much lower effective LR below 10 sr caused by specular reflection. These results are close to our parametrisation of effective LR ($S'_{\text{CALIPSO}}$). In the current version of the lidar simulator we do not parametrise horizontally oriented plates, but in a future version they could be taken into account by parametrising their concentration based on temperature (Noel and Chepfer, 2010). For the ALCs we use the same constant value of the multiple scattering coefficient $\eta = 0.7$ as for liquid cloud droplets (Sect. 4.5).

## 4.4 Cloud overlap and cloud fraction

Model cloud is defined by the liquid and ice mass mixing ratio and cloud fraction in each atmospheric layer. The lidar simulator simulates radiation passing vertically at a random location within the grid cell. Therefore, it is necessary to generate a random vertical cloud overlap based on the cloud fraction in each layer, as the overlap is not defined explicitly in the model output. Two common methods of generating overlap are the random and maximum–random overlap (Geleyn and Hollingsworth, 1979). In the random overlap method, each layer is either cloudy or clear with a probability given by CF, independent of other layers. The maximum–random overlap assumes that adjacent layers with non-zero CF are maximally overlapped, whereas layers separated by zero CF layers are randomly overlapped. COSP implements cloud overlap generation in the Subgrid Cloud Overlap Profile Sampler (SCOPS) (Klein and Jakob, 1999; Webb et al., 2001; Chepfer et al., 2008). The ALC lidar simulator uses SCOPS to generate 10 random subcolumns for each profile, using the maximum–random overlap assumption as the default setting of a user-configurable option. The attenuated volume backscattering coefficient profile and cloud occurrence can be plotted for any

subcolumn. Due to the random nature of the overlap, the attenuated volume backscattering coefficient profile may differ from the observed profile even if the model is correct in its cloud simulation. The random overlap generation should, however, result in unbiased cloud statistics.

## 4.5 Multiple scattering

Due to a finite FOV of the lidar receiver, a fraction of the laser radiation scattered forward will remain in FOV. Therefore, the effective attenuation is smaller than calculated with the assumption that all but the backscattered radiation is removed from FOV and cannot reach the receiver. The forward scattering can be repeated multiple times before a fraction of the radiation is backscattered, eventually reaching the receiver. To account for this multiple scattering effect, the COSP lidar simulator uses a multiple scattering correction coefficient $\eta$, by which the volume scattering coefficient is multiplied before calculating the layer optical thickness (Chiriaco et al., 2006; Chepfer et al., 2007, 2008). The theoretical value of $\eta$ is between 0 and 1 and depends on the receiver FOV and optical properties of the cloud. For CALIOP at $\lambda = 532$ nm a value of 0.7 is used in the COSP lidar simulator. Hogan (2006) implemented fast approximate multiple scattering code. This code has recently been used by Hopkin et al. (2019) in their ceilometer calibration method. They noted that $\eta$ is usually between 0.7 and 0.85 for wavelengths between 905 and 1064 nm. The ALC simulator presented here does not use an explicit calculation of $\eta$, but retains the value of $\eta = 0.7$ for cloud droplets. The code of Hogan (2006) "Multiscatter" is publicly available (http://www.met.reading.ac.uk/clouds/multiscatter/) and could be used in a later version of the framework to improve the accuracy of simulated attenuation and calibration.

## 5 Lidar data processing

Scheme in Fig. 1 outlines the processing done in the framework. The individual processing steps are described below.

### 5.1 Noise and subsampling

ALC signal reception is affected by a number of sources of noise such as sunlight and electronic noise (Kotthaus et al., 2016). Range-independent noise can be removed by assuming that the attenuated volume backscattering coefficient at the highest range gate is dominated by noise. This is true if the highest range is not affected by clouds, aerosol, and if contributions from molecular scattering are negligible. The supported instruments have a range of approximately 8 (CL31), 15 (CL51, CHM 15k) and 30 km (MiniMPL). By assuming the distribution of noise at the highest level is approximately normal, the mean and standard deviation can be calculated from a sample over a period of time such as 5 minutes, which is short enough to assume the noise is constant over this period, and long enough to achieve accurate estimates of the standard deviation. The mean and standard deviation can then be scaled by the square of the range to estimate the distribution of range independent noise at each range bin. By subtracting the noise mean from the measured attenuated volume backscattering coefficient we get the expected attenuated volume backscattering coefficient. The result of the noise removal algorithm is the expected attenuated volume backscattering coefficient and its standard deviation at each range bin.

**Table 5.** Theoretical molecular volume backscattering coefficient calculated at pressure 1000 hPa and temperature 20 °C and the calibration coefficient, relative to the instrument native units, determined for the instrument based on the molecular volume backscattering coefficient and stratocumulus lidar ratio calibration methods.

| Instrument | Wavelength (nm) | Molecular volume backscattering coefficient ($\times 10^{-6} \mathrm{m}^{-1} \mathrm{sr}^{-1}$) | Calibration coefficient |
| --- | --- | --- | --- |
| CHM 15k | 1064 | 0.0906 | 0.34 |
| CL31 | 910 | 0.172 | $1.45 \times 10^{-3}$ |
| CL51 | 910 | 0.172 | $1.2 \times 10^{-3}$ |
| MiniMPL | 532 | 1.54 | $3.75 \times 10^{-6}$ |

## 5.2 Backscatter calibration

ALCs often report attenuated volume backscattering coefficient in arbitrary units (a.u.) or as NRB (MiniMPL). If they report it in units of $\mathrm{m}^{-1} \mathrm{sr}^{-1}$, these values are often not calibrated to represent the true absolute attenuated volume backscattering coefficient. Assuming that range-dependent corrections (overlap, dead time and afterpulse) have been applied on attenuated volume backscattering coefficient in a. u., the reported attenuated volume backscattering coefficient is proportional to the true attenuated volume backscattering coefficient (inclusive of noise backscattering). In order to have a comparable quantity to the lidar simulator and consistent input to the subsequent processing (e.g. cloud detection), calibration by multiplying by a calibration coefficient is required. Formally, the units of the calibration coefficient depend on the units of backscattering recorded by the instrument, which are $\mathrm{m}^{-1} \mathrm{sr}^{-1}$ in CL31, CL51, unitless in CHM 15k and $\mathrm{count.\mu s}^{-1} \mathrm{\mu J}^{-1} \mathrm{km}^{2}$ in MiniMPL, i.e. the units of the calibration coefficient are $\mathrm{m}^{-1} \mathrm{sr}^{-1}$/(instrument units). In the following discussion, we leave out the units. Several methods of calibration have been described previously: calibration based on LR in fully attenuating liquid stratocumulus clouds (O'Connor et al., 2004; Hopkin et al., 2019), calibration based on molecular backscattering (Wiegner et al., 2014) and calibration based on a high spectral resolution lidar reference (Heese et al., 2010; Jin et al., 2015). In addition, calibration can be assisted by sunphotometer or radiosonde measurements (Wiegner et al., 2014).

A relatively large variability of the calibration coefficient has been determined for instruments of the same model (Hopkin et al., 2019). However, past studies can be useful for determining an approximate value of the coefficient before applying one of the calibration methods. For the CL51, Jin et al. (2015) reported a value of 1.2±0.1 based on a multi-wavelength lidar reference. Hopkin et al. (2019) reported mean values 1.4–1.5 for a number of CL31 instruments (software version 202). For CHM 15k, Hopkin et al. (2019) reported mean values between 0.3 and 0.8 for a majority of the instruments examined. The ALCF provides per-instrument default values of the calibration coefficient (Table 5), but a unit-specific coefficient should be determined for an analysed instrument during the lidar data processing step.

Calibration based on LR in fully opaque liquid stratocumulus clouds has been applied successfully on large networks of ALCs. It utilises the fact that given suitable conditions vertically integrated attenuated volume backscattering coefficient is proportional to LR of the cloud, which can be theoretically derived if the cloud droplet effective radius can be assumed. The theoretically derived value is about 18.8 sr for common ALC wavelengths and a relatively large range of effective radii

(O'Connor et al., 2004). Another factor which needs to be known or assumed is the multiple scattering coefficient, which tends to be about 0.7-1.0 in common ALCs. Due to its relatively simple requirements, this method is possibly the easiest ALC calibration method. The ALCF implements this calibration method by letting the user identify time periods with fully opaque liquid stratocumulus cloud, for which the mean LR is calculated. The ratio of the observed LR and the theoretical LR is equivalent to the calibration coefficient. This implementation, while very easy to perform, has multiple limitations, some of which are highlighted by Hopkin et al. (2019):

1. Aerosol can cause additional attenuation and scattering, which results in LR which is different from the theoretical value by an unknown factor. Therefore, a frequent re-calibration may be necessary.

2. The multiple scattering coefficient assumption may not be accurate for the given instrument.

3. The 910 nm wavelength of CL31 and CL51 is affected by water vapour absorption which causes additional attenuation, which is currently not taken into account in the calculation of LR.

4. Near-range attenuated volume backscattering coefficient retrieval is affected by receiver saturation and incomplete overlap. Therefore, using stratocumulus clouds above approximately 2 km for this calibration method is recommended. This range is instrument dependent.

5. The composition of the stratocumulus cloud may be uncertain. At temperature between 0 and -30°C these clouds may contain both liquid and ice which results in a different LR than expected.

These limitations could be addressed in the future by (1) using sunphotometer observations as an optional input to determine the aerosol optical depth (AOD), (2) calculating the multiple scattering coefficient more accurately (such as with the Multiscatter package of Hogan (2006)), (3) calculating the water vapour absorption explicitly based on water vapour, temperature and pressure fields from a reanalysis or radiosonde profile data, (4) correcting the near-range backscatter based on the integrated attenuated volume backscattering coefficient distribution as a function of height of the maximum backscatter (Hopkin et al., 2019, Sect. 5.1), (5) combining the attenuated volume backscattering coefficient profile with temperature field from a reanalysis to exclude cold clouds.

Molecular (Rayleigh) backscattering can be accurately calculated if temperature and pressure of the atmospheric profile is known (Sect. 4.2). This can be employed for absolute calibration of ALCs. Given the low SNR of low-power ALCs, several hours of integration are required to identify the molecular backscattering (Wiegner et al., 2014). The molecular backscattering is attenuated by an unknown amount of aerosol with unknown LR, and the near-range backscattering is affected by a potentially inaccurate overlap correction. Therefore, this method alone produces calibration coefficient which depend on the atmospheric conditions. We found that all studied ALCs except for the CL31 are capable of observing the molecular backscattering (Section 7). Therefore, this method may be used in addition to the liquid stratocumulus LR method for cross-validation of the calibration.

## 5.3 Cloud detection

Cloud is the most strongly attenuating feature in ALC attenuated volume backscattering coefficient measurements. Due to this attenuation, the lidar signal is quickly attenuated in thick cloud and can fall below the noise level before reaching the top of the cloud. This means that the first cloud base can be detected reliably (unless the cloud is too thin or too high and obscured by noise), while the cloud top or multi-layer cloud cannot be observed reliably under all conditions. The opposite is true for spaceborne lidars, which can detect the cloud top reliably but cannot always detect the cloud base. Therefore, ALC observations can be regarded as complementary to spaceborne lidar observations. By applying a suitable algorithm, one can detect CBH, CTH and identify cloud layers. Instrument firmware often determines CBH and sometimes cloud layers as part of its internal processing, often using an undisclosed algorithm which is not comparable between different instruments and potentially not even different versions of the instrument firmware (Kotthaus et al., 2016). Mattis et al. (2016) compared a large number of ALCs and found differences of up to 70 m between the reported CBH, and others found relatively large differences as well (Liu et al., 2015b; Silber et al., 2018). Alternatively to the instrument reported CBH and cloud layers, it is possible to detect cloud based on the attenuated volume backscattering coefficient profile. A relatively large number of cloud detection algorithms have been proposed (Wang and Sassen, 2001; Morille et al., 2007; Martucci et al., 2010; Van Tricht et al., 2014; Silber et al., 2018; Cromwell and Flynn, 2019). We use a simple algorithm based on an attenuated volume backscattering coefficient threshold applied on the denoised backscatter, assuming that the noise can be represented by a normal distribution at the highest range, which is unlikely to contain cloud or aerosol if the instrument is pointing vertically (this may not be true, however, for CL31 which has a maximum range of just 7.7 km). This assumption neglects the range-dependent molecular backscattering, which is relatively small at the ceilometer wavelengths examined (910 nm and 1064 nm). A cloud mask is determined positive where attenuated volume backscattering coefficient is greater than a chosen threshold plus 5 standard deviations of noise at the given range. In addition, the observed attenuated volume backscattering coefficient can optionally be coupled with simulated attenuated molecular volume backscattering coefficient and molecular backscattering removed from the observed backscattering prior to cloud detection. This improves the results in the boundary layer, especially with instruments which operate in the visible range and therefore affected by large molecular backscattering (MiniMPL). A threshold of 2 $\times 10^{-6} \mathrm{m}^{-1} \mathrm{sr}^{-1}$ was found to be a good compromise between false detection and misses in our Southern Hemisphere data relatively unaffected by anthropogenic aerosol. Our observed and simulated results show that cloud backscatter is generally higher than 1 $\times 10^{-6} \mathrm{m}^{-1} \mathrm{sr}^{-1}$, and a threshold below 2 $\times 10^{-6} \mathrm{m}^{-1} \mathrm{sr}^{-1}$ results in excessive false detection due to aerosol, molecular backscattering and noise from sunlight. The threshold is an adjustable option of the ALCF. Users are encouraged to change this value if, for example, they data are affected by a large amount of aerosol. This value is above the maximum molecular backscattering, which is approximately $1.54 \times 10^{-6} \mathrm{m}^{-1} \mathrm{sr}^{-1}$ at the surface in the case of the MiniMPL (wavelength 532 nm). Noise is not simulated by the lidar simulator, but the cloud detection algorithm allows for coupling of simulated and observed profiles, whereby the noise standard deviation is taken from the corresponding location in the observed profile. With 5 minute averaging, when the standard deviation of noise is relatively low, we found that the coupling does not make substantial differences to the detected cloud (not shown). While the threshold-based algorithm is less sophisticated than other methods of

**Table 6.** Location of sites and instruments. The time periods are inclusive.

| Site | Coordinates | Surface altitude (m) | Instruments | Time period | Missing period | Days |
|------|-------------|---------------------|-------------|-------------|----------------|------|
| Cass | 43.0346°S 171.7594°E | 577 | CL51 | 19 Sep–1 Oct 2014 | | 13 |
| Lauder | 45.0379°S 169.6831°E | 370 | MiniMPL, CL31 | 12–24 Jan 2018 | | 13 |
| Christchurch | 43.5225°S 172.5841°E | 45 | MiniMPL, CHM 15k | 17 July–18 August 2019 | 22–31 July | 23 |

cloud detection, the vertical resolution of the simulated attenuated volume backscattering coefficient is likely too low and the vertical derivatives of the simulated attenuated volume backscattering coefficient too crudely represented (Table 7) to apply any algorithm based on the vertical derivatives of attenuated volume backscattering coefficient. Using the same cloud detection algorithm on the observed and simulated attenuated volume backscattering coefficient is essential for an unbiased one-to-one
comparison of cloud.

## 5.4  Water vapour absorption

Previous studies have noted that ceilometers which utilise the wavelength of 910 nm such as the Vaisala CL31 and CL51 are affected by additional absorption of laser radiation by water vapour (Wiegner and Gasteiger, 2015; Wiegner et al., 2019; Hopkin et al., 2019). The wavelength coincides with water vapour absorption bands between 900 and 930 nm, while the other
common ceilometer wavelength of 1064 nm is not affected. Wiegner and Gasteiger (2015) reported that it can cause absorption of the order of 20% in the extratropics and 50% in the tropics. The lidar simulator does not currently account for this. However, as the water vapour concentration is available from the reanalyses and models, it should be possible to use a line-by-line model to calculate the water vapour volume absorption coefficient for each vertical layer during the integration process. Water vapour also affects calibration of the observed attenuated volume backscattering coefficient. In order to use the liquid stratocumulus
LR calibration method, attenuated volume backscattering coefficient has to be corrected for water vapour absorption to achieve high accuracy of calibration. Hopkin et al. (2019) used a simplified approach based on a parametrised curve and reported a difference from explicit radiative transfer calculations of 2% in the United Kingdom atmosphere (Middle Wallop). In the future either approach should be used to include water vapour absorption in the simulator, or remove the effect of water vapour absorption from the observed lidar attenuated volume backscattering coefficient to achieve an improved one-to-one comparison
between the observations, reanalyses and models.

## 6  Description of case studies

The case studies analysed here were selected to include all instruments supported by the framework. We compare four different instruments (CHM 15k, CL31, CL51, MiniMPL) deployed at three locations in NZ (Lauder, Christchurch, Cass) with three reanalyses (MERRA-2, ERA5, JRA-55), one NWP model (AMPS) and one GCM (UM). These case studies aim to demonstrate
capability rather than to comprehensively evaluate cloud simulation in the models and reanalyses. The work detailed in Kuma

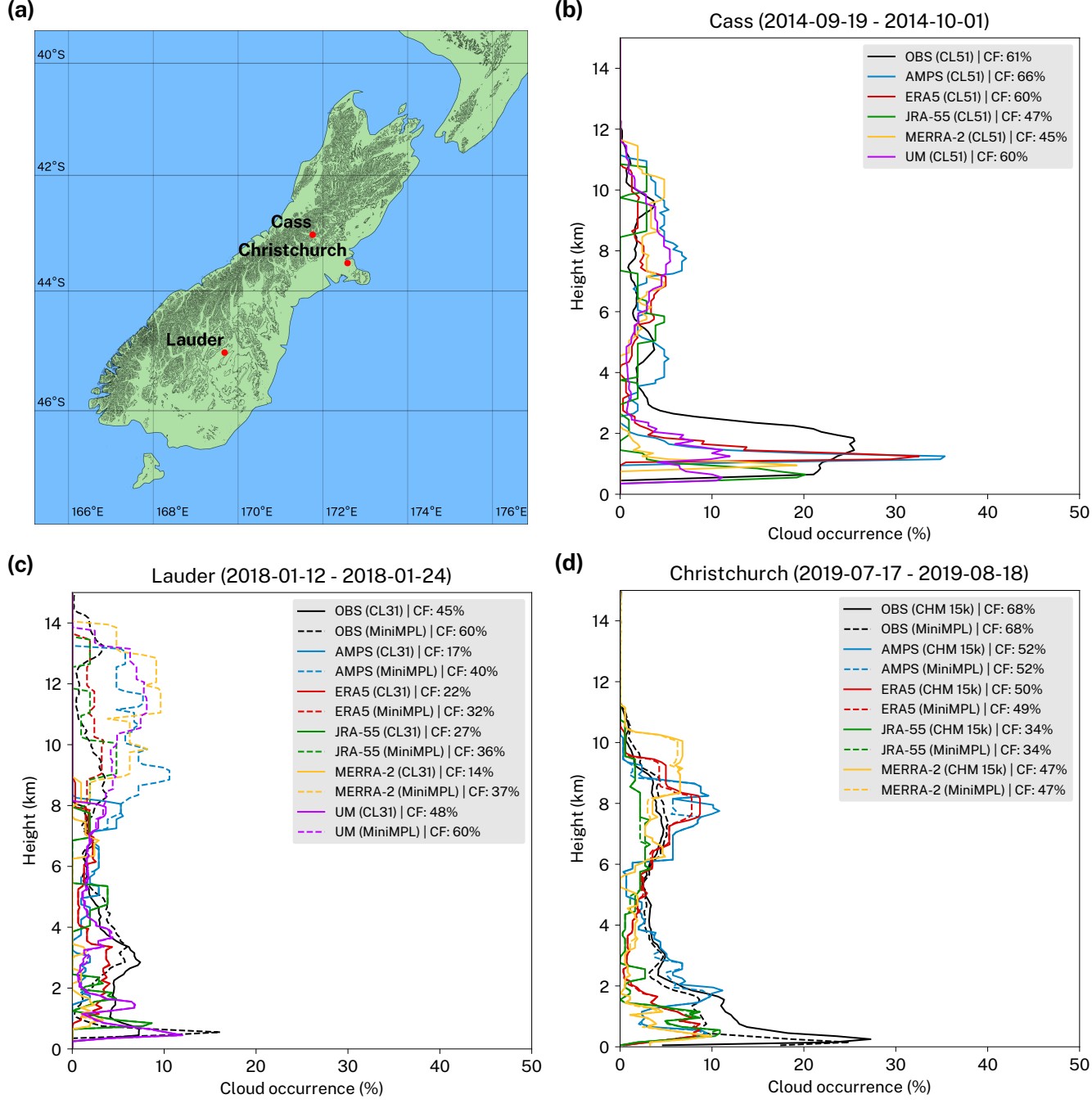

**Figure 3. (a)** Map showing the location of sites. Data at three sites in New Zealand were analysed: Cass, Lauder and Christchurch. **(b), (c), (d)** Cloud occurrence histograms as a function of height above the mean sea level observed at three sites and simulated by the lidar simulator based on atmospheric fields for five reanalyses and models. Shown is also the total cloud fraction (CF). The histogram is calculated from the cloud mask as determined by the cloud detection algorithm.

**Table 7.** Number of models levels and vertical resolution in the range of the instrument at the locations of the case studies. First number is the number of levels, followed by the minimum and maximum distance range between adjacent model levels in the lidar's range (m).

|          | Cass (CL51)    | Lauder (CL31) | Lauder (MiniMPL) | Christchurch (CHM 15k) | Christchurch (MiniMPL) |
|----------|----------------|---------------|------------------|------------------------|------------------------|
| AMPS     | 42; 33–778     | 31; 35–528    | 59; 35–1021      | 43; 33–779             | 60; 33–870             |
| ERA5     | 23; 222–1469   | 17; 220–950   | 30; 220–4748     | 25; 213–1425           | 31; 213–4107           |
| JRA-55   | 23; 223–1479   | 17; 217–948   | 26; 217–1402     | 25; 213–1426           | 26; 213–1426           |
| MERRA-2  | 34; 118–1080   | 26; 125–669   | 47; 125–1329     | 34; 124–1059           | 48; 124–1167           |
| UM       | 44; 70–645     | 33; 32–449    | 65; 32–1181      |                        |                        |

et al. (2020) provides a detailed evaluation of the UM and MERRA-2 relative to shipborne ceilometer observations. Figure 3a shows the location of the sites and Table 6 summarises the case studies, which are also described in greater detail below. The sites were chosen from available datasets to demonstrate the use of the framework with all supported instruments. Two of the sites also had co-located instruments: CL31 and MiniMPL in Lauder, and CHM 15k and MiniMPL in Christchurch. The MiniMPL in Lauder and Christchurch were two different units. The number of models levels within the range of each instrument and vertical resolution range are listed in Table 7.

Cass is a field station of the University of Canterbury located at an altitude of 577 m in the Southern Alps of the South Island of NZ. The station is located far from any settlements and likely affected little by anthropogenic aerosol relative to the other sites. We have analysed 13 days of observations with a CL51 at this station performed in September and October 2014.

Lauder is a field station of NIWA located inland in the Central Otago region on the South Island of NZ. The station is situated in a rural area relatively far from large human settlements at an altitude of 370 m. We have analysed 13 days of co-located MiniMPL and CL31 observations made in January 2018. The MiniMPL was operated in an enclosure with a scanning head set to a fixed vertical scanning mode during this period (elevation angle 90°).

Observations at the Christchurch site were performed at the University of Canterbury campus on the Ernest Rutherford building rooftop at an altitude of 45 m. Christchurch is located on the east coast of the South Island of NZ. Its climate is affected by the ocean, its proximity to the hilly area of the Banks Peninsula, the Canterbury Plains and föhn-type winds (Canterbury northwester) resulting from its position on the lee side of the Southern Alps. The city is affected by significant wintertime air pollution from domestic wood burning and transport. The orography of the city and the adjacent Canterbury Plains is very flat, making it prone to inversions. The Ernest Rutherford building is a 5 floor building situated in an urban area, surrounded by multiple buildings of similar height. We have analysed 23 days of co-located MiniMPL and CHM 15k observations performed in July and August 2019. The MiniMPL was operated in an enclosure with a scanning head set to a fixed vertical scanning mode (elevation angle 90°). The nudged run of the UM was only available up to year 2018. Therefore, it was not analysed for this site.

## 7 Results

To demonstrate the ways that the ALCF can be used we compared a total of 49 days of ALC observations with simulated lidar attenuated volume backscattering coefficient at three sites in NZ (Sect. 6). The observed attenuated volume backscattering coefficient was normalised to calibrated absolute range-corrected attenuated volume backscattering coefficient. The noise mean as determined at the furthest range was removed from attenuated volume backscattering coefficient. Cloud detection based on an attenuated absolute volume backscattering coefficient threshold of $2 \times 10^{-6} \mathrm{m}^{-1}\mathrm{sr}^{-1}$, after removing molecular backscattering and 5 noise standard deviations, was applied to derive a cloud mask and CBH. We compare the statistical cloud occurrence as a function of height above the mean sea level (ASL) (Fig. 3b, c, d) and individual attenuated volume backscattering coefficient profiles (selected profiles are shown in Fig. 4, 5 and 6) in this section. In these plots 5 standard deviations of the attenuated volume backscattering coefficient noise (Sect. 5.3) were removed. In addition, molecular backscattering was removed by coupling the observed data (Fig. 4a, 5a, 6a) with molecular attenuated volume backscattering coefficient calculated by the lidar simulator based the MERRA-2 reanalysis data. The same applies to model data (Fig. 4b–f, 5b–f, 6b–e), but molecular attenuated volume backscattering coefficient was calculated by the lidar simulator based the respective model data.

### 7.1 Cass

We analysed 13 days of CL51 observations at the Cass field station in late winter. Due to the location of the station at a relatively high altitude in a varied terrain of the Southern Alps, the models with their relatively coarse horizontal grid resolution do not represent the terrain and position accurately. The orography representation of the models meant that the virtual altitude of the station was 1115 m (AMPS), 1051 m (ERA5), 401 m (JRA-55), 914 m (MERRA-2) and 428 m (UM). The virtual position, which is the centre of the nearest model grid cell to the site location, ranged from relatively close in the Southern Alps (AMPS, ERA5, MERRA-2, UM) to relatively far on the West Coast of NZ (JRA-55) depending on the horizontal resolution of the grid. The time period examined was characterised by diverse cloud occurrence with periods of low cloud and precipitation, mid-level cloud, fog, high cloud and clear skies. Precipitation, currently not simulated by the lidar simulator, was present in about 18% of the observed attenuated volume backscattering coefficient profiles, as determined by visual inspection. Figure 3b shows that predominantly low cloud and precipitation between the ground and 3 km ASL in 25% of profiles was observed. Cloud between 3 and 12 km ASL was observed about evenly in 2% of profiles. While the reanalyses and models were able to partially reproduce the peak of cloud occurrence near 1 km ASL, the peak they displayed is less vertically broad than observed, and in the UM the peak was much weaker than observed. The lack of precipitation simulation might also have contributed to this apparent difference between observed and simulated cloud. Above 3 km ASL, the reanalyses and models tended to overestimate cloud, with only ERA5 and JRA-55 simulating close to the observed cloud occurrence. The observed total CF was 61%. AMPS overestimated this value by 5 percentage points (pp), ERA5 and the UM reproduced almost the exact value (within 1 pp), while the other reanalyses (JRA-55 and MERRA-2) underestimated CF by about 15 pp.

# Cass (2014-09-25)

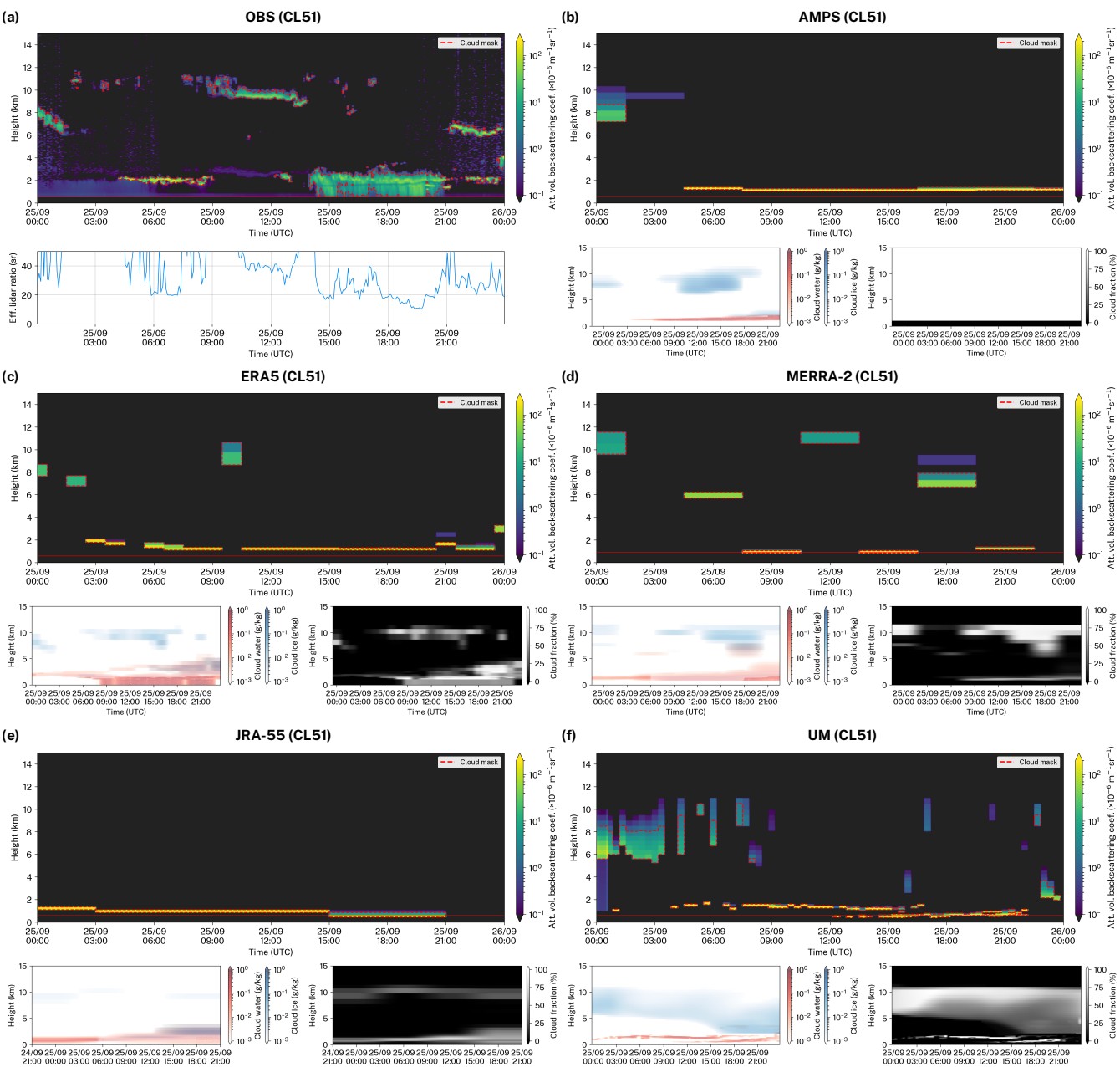

**Figure 4.** Examples of observed and simulated attenuated volume backscattering coefficient during 24 hours at Cass. The observed attenuated volume backscattering coefficient was normalised to absolute units and denoised. The first subcolumn generated by the Subgrid Cloud Overlap Profile Sampler (SCOPS) was used to make the plots. The red line is the station altitude. Shown is also **(a)** observed effective lidar ratio calculated by vertically integrating attenuated volume backscattering coefficient and **(b–f)** the corresponding model cloud liquid water, cloud ice and and cloud fraction.

# Lauder (2018-01-16)

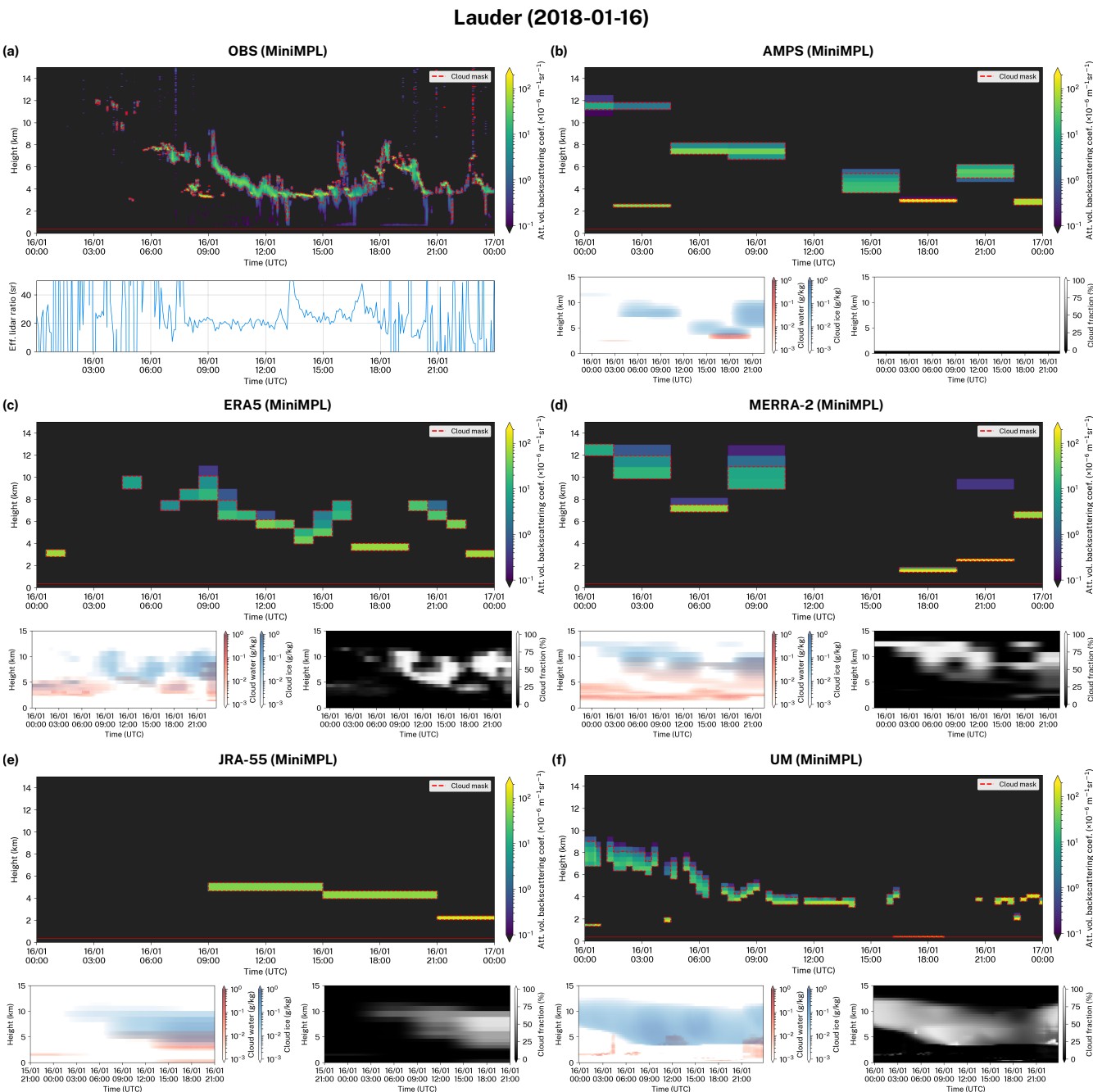

**Figure 5.** The same as Fig. 4 but for the Lauder.

# Christchurch (2019-07-18)

**Figure 6.** The same as Fig. 4 but for the Christchurch.

## 7.2 Lauder

We also analysed 13 days of CL31 and MiniMPL observations at the Lauder station in summer. During the time period relatively diverse cloud was observed, with periods of low, mid- and high cloud, clear sky and a small fraction of profiles with precipitation (about 3%). The altitude of the station of 370 m ASL generally had a much higher equivalent in the reanalyses and models: 565 m (AMPS), 642 m (ERA5), 681 m (JRA-55) and 786 m (MERRA-2) due to the presence of hills in the surrounding region (the station is in a high valley), with the exception of the UM where the altitude was 385 m. The virtual station position in the reanalyses and models ranged from relatively close to the station in the same geographical region (AMPS, ERA5), a nearby location in a more hilly region (JRA-55), a relatively distant location in the adjacent Dunstan Mountains (MERRA-2) and a relatively distant location in Central Otago (UM). Figure 3c shows that the CL31 observed relatively even cloud occurrence between the ground and 3 km ASL at 8%, falling off to about 3% between 4 and 8 km ASL (the maximum lidar range of CL31 is 7.7 km). The MiniMPL observed much weaker attenuated volume backscattering coefficient than CL31 below 3 km ASL, which was identified as an overlap calibration issue in the MiniMPL. The MiniMPL observed substantial amounts of cloud above 8 km, not present in the CL31 observations due to its range limitation. Overall, the observed cloud occurrence had two peaks at ground to 3 km ASL and at about 9 km ASL. The simulated cloud occurrence was generally underestimated between the ground and 5 km ASL, with the exception of the UM which reproduced the lower half of the peak accurately, and ERA5 which reproduced the upper half of the peak accurately. Above 5 km ASL, the cloud occurrence was well reproduced in ERA5 and JRA-55, and strongly overestimated in AMPS, MERRA-2 and the UM. The reanalyses and models also tended to have two peaks at about 2 km ASL and 11 km ASL, but these were quite different from the observed peaks, with the lower peak underestimated by about 5 pp in the reanalyses and models and the higher peak overestimated by about 5–10 pp. The total CF was observed as 45% and 60% by CL31 and MiniMPL, respectively. CF observed by the MiniMPL was likely higher due to its higher maximum lidar range (CL31 missed a substantial amounts of high cloud due to this limitation). The total CF was strongly underestimated by the reanalyses and models by up to 31 pp (CL31) and 28 pp (MiniMPL), with the exception of the UM which simulated the correct CF within 3 pp.

## 7.3 Christchurch

The Christchurch observations were taken during a total of 23 days in mid- to late winter. The cloud situations were characterised by the frequent occurrence of low cloud and fog, with relatively diverse mid- and high level cloud and periods of clear sky also present (not shown). Precipitation was present in about 9% of profiles and fog in about 11% of profiles. As the site location is relatively flat (Canterbury Plains), the models did not have any difficulty in reproducing the altitude of the site, which was 32 m (AMPS), 72 m (ERA5), 143 m (JRA-55) and 76 m (MERRA-2). The virtual location was within the boundaries of the city (AMPS), on the Canterbury Plains close to the city boundaries (ERA5, MERRA-2), and over Lake Ellesmere about 20 km from the city (JRA-55). Figure 3d shows that the co-located CHM 15k and MiniMPL observed a strong peak of cloud occurrence of 26% (CHM 15k) at about 500 m ASL. This was likely due to the combined precipitation and fog as well as false detection of aerosol as cloud. The observed cloud occurrence had a local minimum of 2% at about 5 km ASL, a secondary peak

of 5% at 7 km ASL, and fell off 0% at 11 km ASL. The CHM 15k and MiniMPL observations showed inconsistencies of up to 4 pp. The reanalyses and models underestimated low cloud by 5–10 pp. With the exception of AMPS, they underestimated mid-level cloud by about 5 pp and represented high cloud relatively accurately. The total CF observed was 68%, while the reanalyses and models strongly underestimated CF by up to 34 pp (JRA-55), with underestimates around 20 pp common.

## 7.4 Backscattering on daily scales

Figures 4, 5, 6 show images of attenuated volume backscattering coefficient for three separate days taken from the three case studies. The selected days represent some of the best-matching profiles and demonstrate how well the reanalyses and models can simulate cloud under favourable conditions. As can be seen in the figures, ERA5 and the UM perform the best in terms of temporal and height accuracy of the simulated cloud (Fig. 4c, 4f, 5c, 5f, 6c). This is likely due to the high output temporal resolution of the UM and ERA5 of 20 min. and 1 h, respectively. The UM and ERA5 were able to represent the relatively fine structure of cloud and to a lesser extent the optical thickness (inferred from the strength of backscattering) of the cloud. Deficiencies, however, are readily identifiable. The low cloud in the UM (Fig. 4f) covers too large area relative to observations (Fig. 4a) and the high cloud has a greater vertical extent in the UM. Likewise, the altocumulus cloud observed in Fig. 5a is shifted by several hours in the UM (Fig. 5f). The stratocumulus and nimbostratus cloud, identified visually based on the attenuated volume backscattering coefficient profiles, in ERA5 (Fig. 4c) is markedly lower than observed (Fig. 4a), as well as optically thicker than in reality. The mid-level cloud in ERA5 (Fig. 5c) was located about 2 km higher than observed (Fig. 5a). Precipitation observed in Fig. 6a towards the end of the analysed period was not present in the ERA5 simulated profile (Fig. 6c) due to lack of precipitation simulation in the current lidar simulator (even though rain and snow specific content is available from the reanalysis). AMPS and MERRA-2 had lower cloud representation accuracy. They managed to capture the overall structure of clouds (Fig. 4b, 4d, 5b, 5d, 6b, 6d), but substantial discrepancies were present, some of which were likely due to the relatively low temporal resolution of 3 h. AMPS has, however, relatively high horizontal grid resolution of 21 km. This demonstrates that other factors in the model than resolution have stronger influence on the quality of cloud simulation. JRA-55 was identified as the last in terms of cloud representation accuracy. JRA-55 has the lowest temporal resolution of the studied reanalyses and models of just 6 h, as well as the lowest horizontal grid resolution of 139 km. Therefore, it cannot be expected to capture any fine details of cloud. In the presented profiles (Fig. 4e, 5e, 6e) one can see that the cloud is only crudely represented. JRA-55 was able to represent the stratocumulus cloud of Fig. 4a, although its temporal extent and optical thickness were overestimated. The mid-level clouds of Fig. 5a and 6a were relatively well-represented in terms of height and optical thickness, given the low temporal resolution of the reanalysis. We stress that a direct attenuated volume backscattering coefficient profile intercomparison is highly dependent on the temporal resolution of the model output. The statistical intercomparison, however, should still give unbiased results if the cloud physics is accurately simulated by the atmospheric model.

Figure 4a, 5a, 6a also show effective LR of observations calculated by integrating vertically attenuated volume backscattering coefficient (Sect. 4.1). If attenuated volume backscattering coefficient is properly calibrated, under fully attenuating cloud conditions effective LR converges to the theoretical value of LR of liquid cloud droplets (approximately 18.8 sr at near IR wavelengths) mutiplied by the multiple scattering coefficient (approximately 0.7; Sect. 4.5).

## 7.5 Molecular backscattering, aerosol backscattering and noise

Figure 7 shows attenuated volume backscattering coefficient histograms as a function of height for small values of the coefficient (up to $2 \times 10^{-6} \mathrm{m}^{-1} \mathrm{sr}^{-1}$) observed and simulated at the sites of the case studies, calculated for the entire time period of each case study. The scale of values is below cloud backscattering, and therefore shows backscattering which results from

molecular and aerosol scattering and noise. Molecular backscattering depends on the atmospheric pressure and temperature as well as the lidar wavelength. It causes the main "streak" (a local maximum) visible in each of the histograms. The observed molecular attenuated volume backscattering coefficient at the surface approximately corresponds to the theoretically calculated value at each wavelength: $0.0906 \times 10^{-6} \mathrm{m}^{-1} \mathrm{sr}^{-1}$ ($\lambda$ = 1064 nm), $0.172 \times 10^{-6} \mathrm{m}^{-1} \mathrm{sr}^{-1}$ ($\lambda$ = 910 nm) and 1.54 $\times 10^{-6} \mathrm{m}^{-1} \mathrm{sr}^{-1}$ ($\lambda$ = 532 nm) at 1000 hPa and 20 °C (Table 5). The molecular backscattering in the boundary layer is, how-

ever, superimposed on backscattering by aerosol and cloud. In the case of the MiniMPL observations at the Christchurch site (Fig. 7i), the molecular attenuated volume backscattering coefficient streak has multiple secondary streaks. These are caused by different levels of attenuation by cloud and aerosol during the period of the observations. These secondary streaks were also partially reproduced by the simulator (Fig. 7j). A smaller portion of the width of the streak is also caused by fluctuations of atmospheric temperature and pressure. Under suitable conditions, molecular attenuated volume backscattering coefficient

can be used for absolute calibration of an instrument. With the exception of CL31 (Fig. 7c), the molecular backscattering can be identified in the observed attenuated volume backscattering coefficient in each case. Therefore, it is possible to choose a calibration coefficient such that the observed and simulated molecular attenuated volume backscattering coefficients overlap. This can be considered a viable alternative to the liquid stratocumulus LR calibration method, or as a means of cross-validating the instrument calibration. However, it should be noted that the accuracy of this method is affected by an unknown amount of

aerosol attenuation. Cloudy profiles can be filtered when calculating the histogram, and therefore the effect of cloud attenuation can be minimised. In addition to the molecular attenuated volume backscattering coefficient streak, there is a zero-centred streak visible in the histograms. This is caused by noise when the signal is fully attenuated by cloud. Lastly, a zero-centred "cone" of noise is visible in the observed attenuated volume backscattering coefficient, increasing with the square of range. The size of this cone is particularly large in the case of the CL31 (Fig. 7c), which is most likely the result of its low receiver

sensitivity and low power compared to the other instruments. The standard deviation of the cone at the furthest range is used to determine the noise standard deviation used by the cloud detection algorithm (Sect. 5.3).

Figure 8 shows the same information as Fig. 7, but for clear sky profiles only. Here, it can be seen that the zero-centred peak caused by the complete attenuation by cloud is no longer present. There is a clear overlap between the centre of the noise cone with the simulated molecular attenuated volume backscattering coefficient; i.e. the noise cone is centred at the

observed molecular attenuated volume backscattering coefficient. This is visible with all instruments including CL31 (Fig. 8c), where the overlap between the observed and simulated molecular attenuated volume backscattering coefficient is most clearly visible at about 1 km ASL. Below 1 km ASL, the effect of boundary layer aerosol distorts molecular attenuated volume backscattering coefficient by an unknown quantity. The clear sky histograms as shown in Fig. 8 may therefore be preferable to the all-sky histograms of Fig. 7 for calibration by fitting molecular attenuated volume backscattering coefficient. The dead

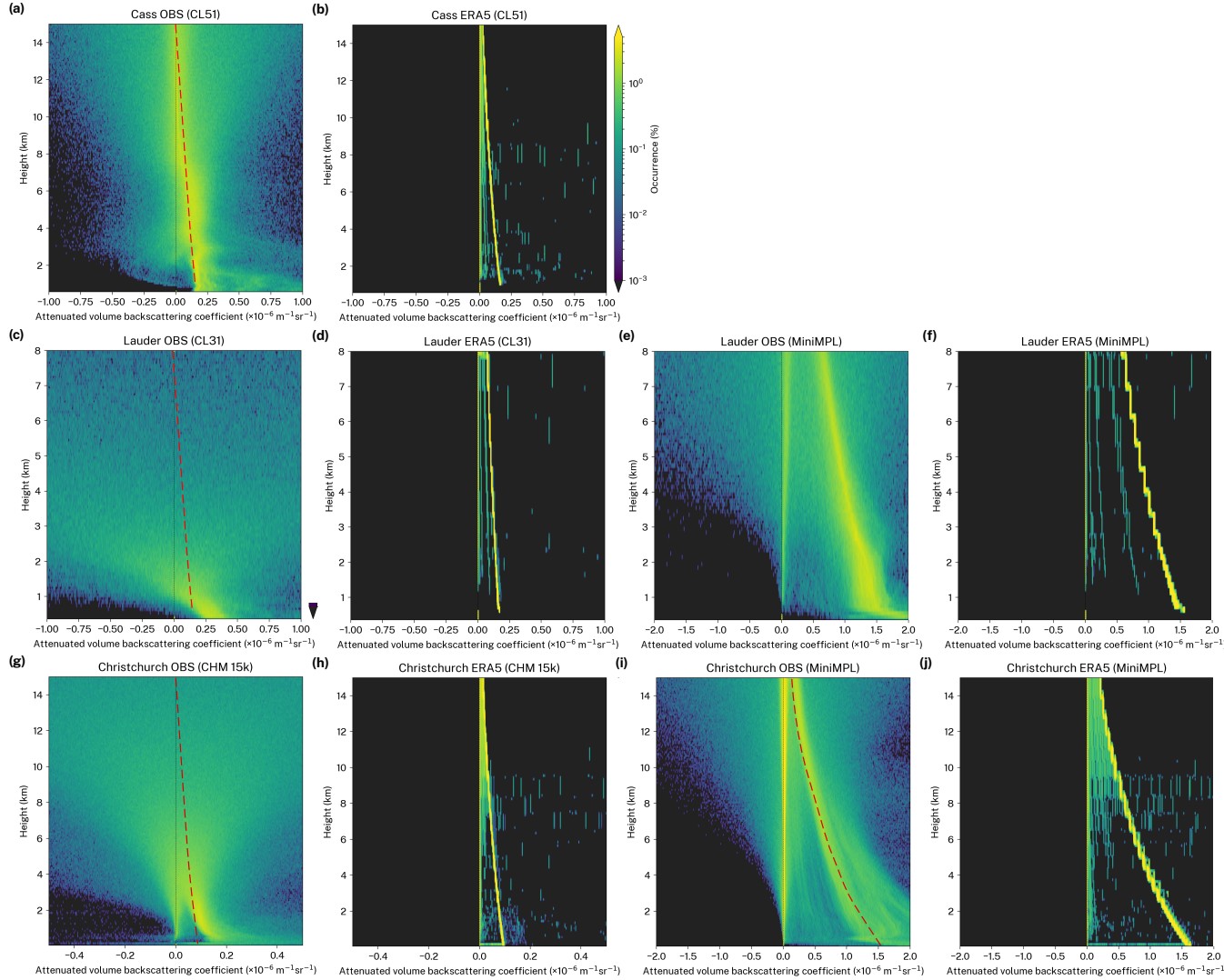

**Figure 7.** Attenuated volume backscattering coefficient histograms as a function of height observed and simulated at three different sites of the case studies calculated from all profiles. The plots show the distribution of attenuated volume backscattering coefficient for values which are on the scale of noise, molecular and aerosol backscattering ([-0.5, 0.5] for CHM 15k, [-1, 1] for CL31 and CL51 and [-2, 2] $\times 10^{-6} m^{-1} sr^{-1}$ for MiniMPL). The simulated attenuated volume backscattering coefficient is based on the ERA5 atmospheric fields. Visible in the plots is backscattering caused by molecular backscattering (the main "streak"), noise when signal is fully attenuated by cloud (the zero-centred "streak"), and the range-dependent noise (the zero-centred "cone"). The molecular backscattering is marked by a red dashed line on the observed attenuated volume backscattering coefficient plots, the shape of which is taken from the simulated molecular attenuated volume backscattering coefficient for the corresponding instrument and site.

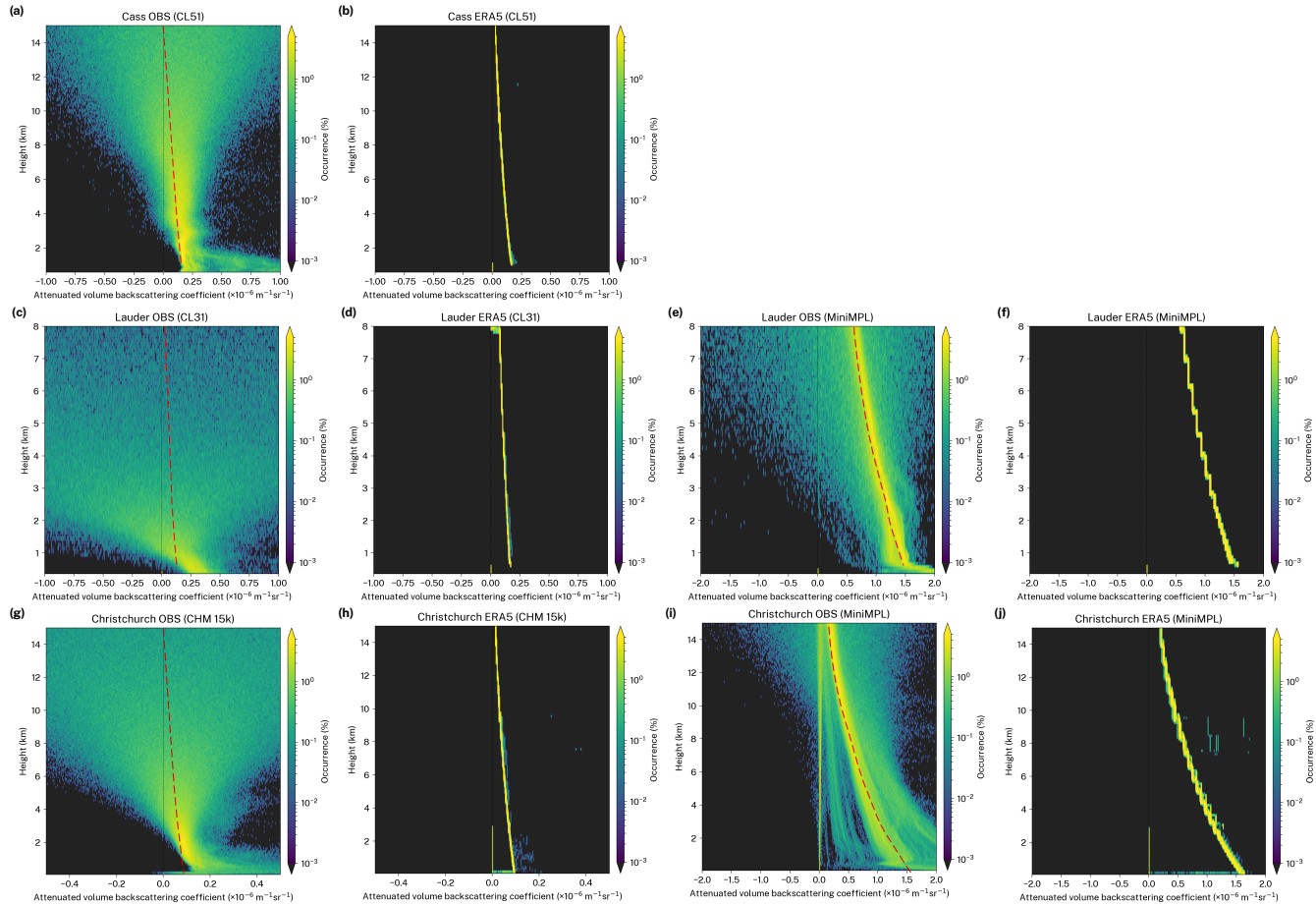

**Figure 8.** The same as Fig. 7 but calculated from clear sky profiles only.

time, afterpulse and overlap MiniMPL calibration supplied by the vendor appears to be deficient and causes range-dependent bias in the attenuated volume backscattering coefficient profile.

We now examine the noise in each instrument using the ALCF. Figure 9 shows the distribution of standard deviation of backscatter noise determined at the highest observable range of each instrument and range-scaled to 8 km. It can be seen that the CL31 is affected by the greatest amount of noise, peaking at about $2 \times 10^{-6} \mathrm{m}^{-1} \mathrm{sr}^{-1}$. This is at the threshold of cloud detection of $2 \times 10^{-6} \mathrm{m}^{-1} \mathrm{sr}^{-1}$. Therefore, thin cloud may be obscured by noise at higher ranges with this instrument. The MiniMPL, operating in the visible spectral range, shows a strongly bimodal distribution of the attenuated volume backscattering coefficient noise depending on sunlight. During daytime, it peaks at about $0.7 \times 10^{-6} \mathrm{m}^{-1} \mathrm{sr}^{-1}$, which is the second highest of the analysed instruments. During nighttime, it peaks at about $0.02 \times 10^{-6} \mathrm{m}^{-1} \mathrm{sr}^{-1}$, which is the lowest of the analysed instruments. The CHM 15k and CL51 peak between the nighttime and daytime MiniMPL at about $0.05 \times 10^{-6} \mathrm{m}^{-1} \mathrm{sr}^{-1}$. All of CL31, CL51 and CHM 15k show a slight reduction of noise during nighttime, presumably because of a small amount of incoming solar radiation

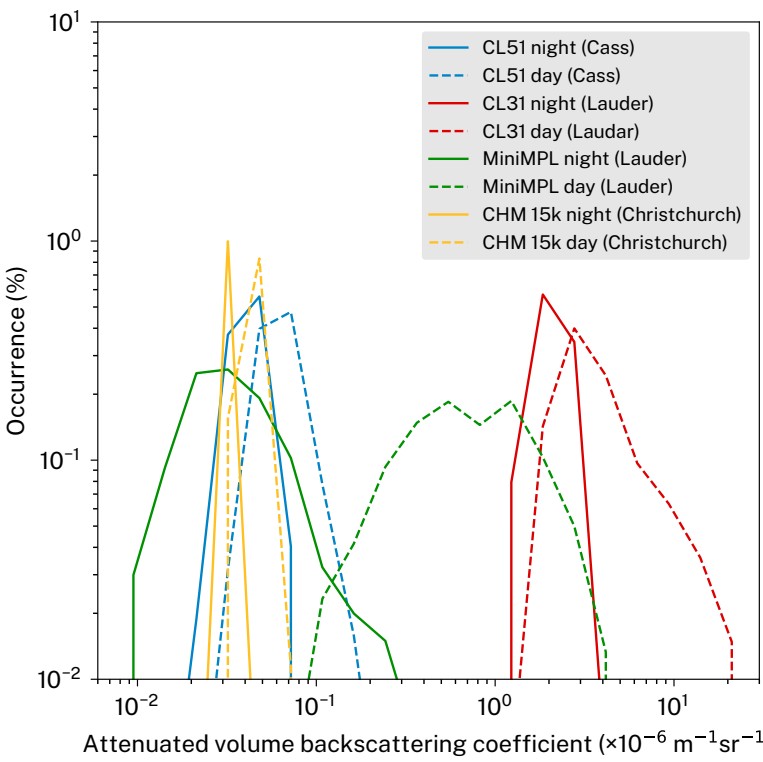

**Figure 9.** Attenuated volume backscattering coefficient noise standard deviation histogram calculated for each instrument at sites of the case studies from clear sky profiles over the whole time period. The noise distribution is calculated at the furthest range. Shown is the range-scaled noise distribution at a range of 8 km. "Night" and "day" distributions are calculated separately from nighttime and daytime profiles only.

at near IR wavelengths. The difference between the nighttime and daytime attenuated volume backscattering coefficient noise in the MiniMPL has been previously analysed by Silber et al. (2018) (Fig. S3) and these results confirm their findings.

## 8 Discussion and conclusions

We presented the Automatic Lidar and Ceilometer Framework, which combines lidar processing and lidar simulation for the purpose of model evaluation. The lidar simulation is based on the COSP spaceborne lidar simulator by accounting for the different geometry and lidar wavelength. We calculated new lookup tables for Mie scattering for a number of ALC wavelengths, developed ice crystal backscattering parametrisation based on temperature and implemented noise removal and cloud detection algorithms. The framework supports the most common ALCs and reanalyses. We demonstrated the use of the framework on ALC observations at three different sites in New Zealand, and applied the lidar simulator to three reanalyses and two models. We found that while some reanalyses and models such as the UM and ERA5 show relatively good correspondence with observed cloud, others performed relatively poorly in our time-limited local comparison. All reanalyses and models

underestimated the total CF by up to 34 pp, with underestimation by 20 pp common. In some cases, the observed and simulated attenuated volume backscattering coefficient profiles matched relatively closely in terms of time and altitude, and a better match was observed with reanalyses with high output temporal resolution such as the UM and ERA5, while reanalyses with low temporal resolution did not allow for reliable direct (non-statistical) comparison of cloud. However, it is clear that more factors than the horizontal and vertical resolution influence the cloud simulation accuracy; especially the cloud, boundary layer and convection schemes employed by the atmospheric model. The reanalysis and model output temporal, horizontal grid resolution and vertical resolution are not always the same as the internal resolution of the underlying atmospheric model. Both have an impact on the comparison between the simulated and observed attenuated volume backscattering coefficient and cloud. While the output resolution should not have an impact on the long-term statistics, it can be a limiting factor for direct attenuated volume backscattering coefficient profile comparison. We demonstrated that the ALCF could be used to identify substantial differences in cloud attenuated volume backscattering coefficient which were present in all reanalyses and models. We showed that all the studied instruments except for the CL31 are capable of detecting molecular backscattering and that this can be used for calibration or for cross-validation of other calibration methods. We found that the nighttime MiniMPL was subject to the least amount of noise of all the instrument examined, followed by the CL51, CHM 15k, daytime MiniMPL and CL31. Noise in the MiniMPL, and to a lesser extent in the other ALCs, was shown to have a bimodal distribution due to day/nighttime. The ALCF can therefore be useful for testing the quality of collected data.

Currently the framework has several limitations which should be addressed in the future. The water vapour absorption at 910 nm likely affects the instrument calibration of the CL31 and CL51 ceilometers and limits the accuracy of the one-to-one comparison, even though due to the relatively high backscattering caused by cloud, the calculated cloud masks are unlikely to be strongly affected. The lidar simulator currently does not simulate backscattering from precipitation. Observed precipitation is generally detected as "cloud" by the cloud detection algorithm, while the simulated profile contains no backscattering at the location of precipitation (backscattering and attenuation by rain drops and snow should be implemented in the lidar simulator in the future). If desired, the attenuated volume backscattering coefficient profiles affected by precipitation can be excluded before the comparison or their fraction determined by visually inspecting the observed attenuated volume backscattering to assess their possible effect on the statistical results. Aerosol is also not currently implemented in the simulator. Previous studies (Chan et al., 2018) characterised optical parameters of different groups of aerosol, which could be used in a future version of the simulator with models which provide concentration of aerosol in their output. In our case studies aerosol volume backscattering coefficient was less than $2 \times 10^{-6} \mathrm{m}^{-1} \mathrm{sr}^{-1}$ and below 4 km, which could result in worst-case two-way attenuation of about 50% assuming LR of 50 sr. This should not preclude cloud detection due to the large magnitude of typical cloud backscattering. The ALCs also suffer from various measurement deficiencies. Notably incomplete overlap, dead time and afterpulse corrections tend to give sub-optimal results at the near range. It is possible to use semi-automated methods to correct for these deficiencies, such as by calculating the integrated attenuated volume backscattering coefficient distribution by height of the maximum backscattering and correcting for the range-dependent bias (Hopkin et al., 2019, Sect. 5.1). This method could be implemented in the framework to enable range-dependent calibration of the observed attenuated volume backscattering coefficient.

The presented framework streamlines lidar data processing and tasks related to lidar simulation and model comparison. The framework was recently used by Kuma et al. (2020) for Southern Ocean model cloud evaluation in the GA7.1 model and MERRA-2 reanalysis. Considering the existing extensive ALC networks worldwide there is a wealth of global data. We therefore think that ALCs should have a greater role in model evaluation. Satellite observations have long been established in this respect due to their availability, spatial and temporal coverage and their well-developed derived products and tools. ALCs, with their diverse formats and decentralised nature, have so far lacked derived products and tools which would make them more accessible for model evaluation. We hope that this software will enable more model evaluation studies based on ALC observations. Development of lidar data processing is currently hampered by closed development of code. We note that code has very rarely been made available with past ALC studies. Continued improvement of publicly available code for lidar data processing is needed to achieve faster development of ground-based remote sensing and make it more attractive for GCM, NWP model and reanalysis evaluation.

*Code and data availability.* The *ALCF* is open source and available at https://alcf-lidar.github.io and as a permanent archive of code and technical documentation on Zenodo at https://doi.org/10.5281/zenodo.4088217. The technical documentation is also in the Supplementary information. A tool for converting Vaisala CL31 and CL51 data files to NetCDF *cl2nc* is open source and available at https://github.com/peterkuma/cl2nc. A tool for converting MiniMPL raw binary data files to NetCDF *mpl2nc* is open source and available at https://github.com/peterkuma/mpl2nc. The observational data used in the case studies are available upon request. The reanalyses data used in the case studies are publicly available online from the respective projects. The *Unified Model* data used in the case studies are available upon request. The Unified Model is proprietary to the UK Met Office and is made available under a licence. For more information, readers are advised to contact the UK Met Office.

*Author contributions.* Peter Kuma wrote the code of the framework, performed the data analysis of the case studies and wrote the text of the manuscript. Adrian McDonald and Olaf Morgenstern provided continuous scientific input on the code development, analysis and text of the manuscript. Richard Querel, Israel Silber and Connor J. Flynn provided calibration of the MiniMPL data and substantial discussion of the theoretical concepts. All authors reviewed the manuscript.

*Competing interests.* The authors declare that they have no competing interests.

*Acknowledgements.* We would like to acknowledge the New Zealand Deep South National Science Challenge project which provided funding for our work; the New Zealand eScience Infrastructure (NeSI) which provided supercomputing resources to run the Unified Model; Vidya Varma, Jonny Williams, Guang Zeng and Wolfgang Hayek for their contribution to setting up a nudged run of the Unified Model; Graeme Plank and Graeme MacDonald who participated on the installation of the Vaisala CL51 at the Cass field station; the COSP project

for the code which we used as the basis for the lidar simulator; the AMPS, JRA-55, ERA5, MERRA-2 models and reanalyses which provided public access to their data; the open source libraries numpy (Van Der Walt et al., 2011), scipy (Virtanen et al., 2019), matplotlib (Hunter, 2007), netCDF4 (Rew and Davis, 1990) and Astropy (Price-Whelan et al., 2018) and the Python programming language (Rossum, 1995) which we used in the implementation of our code; the R programming language (R Core Team, 2017), the Natural Earth dataset (https://www.naturalearthdata.com) and the Shuttle Radar Topography Mission (SRTM) Version 3 Global 1 arc second digital elevation model (Werner, 2001; NASA JPL, 2013) which we used to produce a map of sites; GitHub which provided free hosting of our code; and the Linux-based (Torvalds, 1997) operating systems Devuan GNU+Linux and Debian GNU/Linux on which we produced this analysis.

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
