# Peer review of "Ground-based lidar processing and simulator framework for comparing models and observations (ALCF 1.0)"

_Geoscientific Model Development, 2020_

## Referee Comment (RC1) · Anonymous Referee #1 · 15 Jul 2020

In recent years automated lidars and ceilometers (ALCs) are increasingly used for many atmospheric studies in particular when vertical profiles of clouds and aerosols are of interest. Many applications concern the determination of cloud bottom heights, mixing layer heights and particle backscatter coefficient profiles. Insofar provision of tools to handle ALC-data are quite useful – Kuma et al.'s paper can constitute a useful contribution.

In their paper they introduce a processing and simulation framework (ALCF1.0): the paper includes a description of the main features, and examples how it can be used for cloud studies. The paper fits to the scope of GMD however, before publication a few

clarifications are required, and the structure must be revised (order of figures should be 1,2,3,..., outline of the paper), it seems that the reorganisation has been done very fast after the "quick review".

**General comments:**

1. The requirements on simulating and/or evaluating ALC-signals depend on the application: for the determination of cloud bottom heights (CBHs) they are certainly quite different compared to particle backscatter coefficients. It should be made very clear in the paper, which application is aimed at and how the requirements for this application are fulfilled. An outlook on planned extensions can be given for some relevant applications not yet implemented.

2. Section 2: The lidar simulator should be explained in more detail and whether the focus of ALCF1.0 is on clouds or aerosols or both. In the present state it seems to be "clouds", as this is much easier to be treated: The numerical models/reanalyses provide the necessary information and the variability of the lidar ratios is comparable small. Low clouds typically consists of droplets so the consideration of non-spherical ice crystals is not highly important (provided that the focus is on CBHs only). Anyway: it is strongly recommended to include to consideration of optical properties of ice crystals into ALCF2.0 (phase functions are available since decades).

   The treatment of aerosols is not sufficiently explained: the backscatter signal depends on the aerosol distribution and their optical properties. Where is this input coming from in cases when the model/reanalysis does not provide aerosol information (NWP not necessarily consider aerosols). What about non-spherical particles – in the case of e.g. dust or volcanic ash the application of Mie theory is certainly not justified. Even if aerosol applications are not included in the paper this topic must be discussed (maybe as outline of the next version of ALCF). Chan et al. (2018) have demonstrated that there is an influence of particle shape

Interactive
comment

on the backscatter profiles (and the intercomparison with model results), and that consideration of aerosols and their non-sphericity is possible.

Chan et al: Evaluation of ECMWF-IFS (version 41R1) operational model forecasts of aerosol transport by using ceilometer network measurements, Geosci. Model Dev., 11, 3807–3831, https://doi.org/10.5194/gmd-11-3807-2018, 2018.

3. Section 4:

In the introduction of this section again the description of the treatment of aerosols has been forgotten. In section 4.1 (starting with the paragraph above Eq. 2) it should clearly stated whether the authors talk of clouds or aerosols. Aerosol size distributions typically are not described by gamma-distributions, and the application of Mie theory is often not adequate.

Moreover, the authors should comment how they deal with the lidar ratio: when CBHs shall be derived the magnitude of the lidar ratio is not very important: the backscatter coefficient is such large that the uncertainty of the lidar ratio is more or less a second order effect. They should also comment on the consequences of the wrong description of ice crystals by the Mie theory: is this tolerable in view of the overall relevance of the lidar ratio in case of CBHs?

Are the derived microphysics of the cloud particles (mainly size distribution) checked against the input used in the models/reanalyses. If it is available it should be consistent (ALCF vs. model).

In Section 4.3 multiple scattering is briefly described. Of course it belongs to a full description of radiative transfer but in case of CBH-determination I don't believe that this is really relevant (the photons stay close to the optical axis). A clarifying sentence might be added.

4. Section 7:

This section is quite long and should be separated into subsections. If done so 7.1 would deal with Fig. 4, Section 7.2 with Figs. 5-7, and Section 7.3 with Fig. 8 and so on. In this context I suggest to delete Section 7.3 (Fig. 8): The arguments of the authors with respect to the cloud albedo are very difficult to understand and thus not convincing, and for me highly speculative: How is the effect of overlapping clouds in different levels considered? How accurate is the estimation of the lidar ratio (ice crystals)? The calculation of the lidar ratio from the vertically integrated backscatter is not explained (in constrast to the extended description of $r_{\text{eff}}$ etc. in Section 4.1). How is obscuration of high clouds by low clouds considered, and so on. An ALC can provide a lot of useful information so it is not necessary to "invent" retrievals of further parameters associated with low confidence.

In connection with Figs. 5–7 it should be outlined why the backscatter coefficient is plotted and not the cloud mask. In the latter case the model will show clouds that might be obscured by low clouds in case of ALC-measurements. This could give further insight into the fundamental differences between remote sensing and modeling (and the problems associated with the resolution).

In connection with Figs. 9–10 again the aerosol related questions from above show up.

**Specific/minor comments:**

In the following $i/j$ means page $i$ line $j$.

1. 2/8: "aerosol optical depth" should be replaced by particle backscatter coefficient: AOD is not a primary output of ALCs as it is an integral quantity, and depends on the lidar ratio.

2. 2/14: "thousands" is a little bit exaggerated: each network rather consists of a few hundreds at maximum.

3. 2/18: "model evaluation": here a few references would be fine.

4. 4/13 ff: check order!

5. 5/18 (and throughout the paper): please avoid terms like "backscatter": either backscatter coefficient, particle backscatter coefficient, attenuated backscatter or any other clearly defined physical quantity.

6. 6/1: "in the near IR spectrum" can be deleted, the same is true for "in the visible spectrum" in line 17.

7. 6/2: the native resolution is indeed 5 m, but the typical output-resolution is 15 m.

8. 6/3: "uncalibrated": this is a little bit misleading as the signals (to my knowledge) are calibrated against a standard instrument. This has been done to make all instruments within a network comparable. I don't know if this applies to the ceilometers in New Zealand as well.

9. 6/18: "is up to 5 m": what does this mean? Is the resolution coarser (10 m or so) or finer (3 m)?

10. 6/19: "The instrument can be housed...": Why is this mentioned? It is true for the other instruments as well.

11. 7/2 (and later): when giving the number of vertical levels it should be added how many levels are within the range of the ALCs, e.g. CL31. And the typical vertical resolution in this range should be mentioned.

12. 7/10: What does "single level" mean?

13. 8/7: What is meant be "horizontally homogeneous"? The lidar equation is one-dimensional (along the pointing of the laser beam, typically vertically).

14. 8/21: "usually lower": I would rather write "much smaller". The backscatter coefficient of (low) clouds is several orders of magnitude larger than Rayleigh scattering. The lidar ratio of Rayleigh-scattering should be mentioned.

15. 12/31 ff: I don't know if it is necessary to consider calibration constant from "past studies". Most ceilometers show calibration constants that are not really constant (temperature-dependent), and maybe some instruments have been set to a higher/lower sensitivity on purpose. If ALCF1.0 offers the option to calibrate signals (Rayleigh or cloud method) it is not required to rely on previous measurements anyway.

16. 14/24: It is certainly a good idea to define a absolute threshold for the cloud detection. However, $2 \times 10^{-6}$ m$^{-1}$ sr$^{-1}$ corresponds to an extinction coefficient of 0.04 km$^{-1}$ (assuming a lidar ratio = 20 sr), I would have expected a larger value. Maybe a few additional comments on the magnitude can be added (typical values for low water clouds).

17. 14/31: "simulated backscatter too crudely": see comment 7/2.

18. 15/25: Only the altitude of Cass is given here. To be consistent the altitude of the other stations should be given as well in this section.

19. 16/12: Typo in unit.

20. 18/3 ff: When discussing Figs. 5–7, the panels of the lidar ratio are ignored. They are also not explained in the figure captions. What is the meaning of these panels? Can they be omitted?

21. 18/5: What is "favourable"? Is it meant that individual intercomparisons of CBH between ALC and models are more or less impossible due to the different spatio-temporal resolution. So the focus must be on climatologies, and the provision of the true CBH and vertical cloud distribution for model-validation.

22. 18/27: Fig. 5l does not exist (typo). More examples like this are existing.

23. 20/32: "relatively poorly". Here it should be stressed that the sample is limited, thus general conclusions might be difficult.

24. 42: Figure caption should be revised: One description of "day and night" is sufficient.

───────────────────────

---

## Referee Comment (RC2) · Anonymous Referee #2 · 17 Jul 2020

Recently, more and more networks of automated lidars and ceilometers (ALC) have been put into operation or have been extended mainly by weather services. These temporal and spatially very densely covered measurements of instruments from different manufacturers provide a valuable data set for the investigation of cloud bottom heights and cloud distribution. ALCs however, can also be used for the determination of the mixing layer height or physical aerosol parameters, i.e. the particle backscatter coefficient. To make use of this large amount of data for model or even satellite instrument validation, an automated way of processing these different data sets is necessary in order to have a harmonized database. In this paper, Peter Kuma et al. provide with ALCF1.0 a framework on an open source basis which addresses this deficiency. The

paper fits within the scope of GMD and needs some minor revisions before publication.

**General comments**

The paper deals primarily with comparing cloud distribution, the cloud base height and cloud layers from observations and models. However, the influence of aerosol on radiative transfer has to be considered in any case if quantitative statements are to be made. For the investigation of only the mentioned cloud parameters a calibration is not absolutely necessary except for the harmonisation of the data and to find a common cloud detection threshold. However, with a view to a next version of the framework, the already included parts such as multiple scattering, Mie and Rayleigh scattering calculations and the calibration itself provide already the basis to include the option of investigating aerosol optical properties as well.

The authors should emphasize that the focus in this version of the framework is on clouds and not on aerosol which has influence on the attenuated backscatter of observations and of simulated lidar signals.

Section 7: This is a rather long section and should be divided into subsections dealing with the different figures 4, 5-7, 8, 9-11.

Fig. 5 it seems that for some reanalyses and models the cloud layer near the ground seem to fully attenuate the signal and no clouds above can be detected whereas higher clouds are visible in the observation. This is especially true for the models and reanalyses with very coarse resolution. However, the reanalyses and models may have calculated the clouds, but they are not visible in the simulated lidar signal. Have you compared the observed signal also with cloud masks or cloud fraction of the models? Is the picture different if you would plot other subcolumns than the first one? The lidar ratio subplots in Fig. 5-7 can be omitted since they don't appear neither in the text nor in the caption.

Fig. 8: Aerosol would have influence on the lidar ratio determined with the applied

calibration method in this paper as stated on P.15 L15. Aerosol in the layers below the cloud would lead to an increase of the determined lidar ratio (decrease of $LR^{-1}$) what can be seen in Fig. 8. Because aerosol is not considered in the used models here, they are not affected. Could you please comment on how you screened the data for aerosol and is its influence somehow considered in your calculations?

**Specific comments**

1. P.2 L.4: ash is particulate matter as well. Thus, just aerosol is sufficient.

2. P.2 L.8: "aerosol optical depth" is an integrated quantitiy relying on the lidar ratio. ALCs are used to determine the particle backscatter coefficient: Change to particle backscatter coefficient

3. P.2 L.15: EARLINET is not a network of ALCs. Most of the instruments are not operated autonomously. The measurements must meet stringent quality control criteria and are performed for selected times. A better example would be PollyNET:

   Baars et al.: An overview of the first decade of PollyNET: an emerging network of automated Raman-polarization lidars for continuous aerosol profiling, https://doi.org/10.5194/acp-16-5111-2016

4. P.6 L.19: "instrument can be housed in a protective enclosure": This is not only true for the MiniMPL systems but also for the other ALCs to allow a 24/7 operation at any location.

5. P.8 L.10: As already mentioned, it should always be emphasized that the focus here is on clouds and that aerosol is not taken into account. This however would be necessary to perform the correct radiative transfer calculations in order to appropiately simulate the lidar signal.

6. P.13 L.1 and 2: Is the calibration constant a unitless quantity?

7. P.14 L.24: The cloud detection threshold of $2\times10^{-6}m^{-1}sr^{-1}$ corresponds to a particle extinction coefficient of 0.1 $km^{-1}$ if assuming a lidar ratio of 50 sr which is not unusual for aerosol. This threshold seems to me rather low and could lead to misclasification of aerosol as clouds. This is even more true in the case of the MiniMPL system which is operated at 532 nm. Could you please comment on the observed magnitudes within liquid and cirrus clouds? Why is no adaption to the wavelength needed? Would it make sense to apply a height dependent threshold with lower values at higher altitudes to account for the lower attenuated backscatter of cirrus clouds?

8. P.16 L.11: Please check the use of the different terms "attenuated backscatter", "total attenuated backscatter", "total volume backscatter coefficient" and "particle backscatter coefficient" in the text, the figure captions and labels. The measured range corrected lidar signal calibrated with the calibration constant is normally referred to as the attenuated backscatter.

9. P.18 L.17, 19, 27 and 29: Please check figure labels for Fig. 5 in the text. There are no subfigures g, h, k, i, l.

10. P.19 L.29: Again, as already mentioned, aerosol can not be neglected. Especially when using the molecular backscatter to determine the calibration coefficient, aerosol must be considered. As you can see in Fig. 9 and 10 the streak caused by molecular backscatter is too broad to retrieve an accurate calibration constant. This is mainly caused by aerosol and varying atmospheric conditions (temperature, pressure). I would omit here and in P.20 L.10 the proposal of a "new" method.

11. P.20 L.9: Fig. 10 instead of Fig. 9

12. P.20 L.21 and 24: You are only addressing the difference between day and night here for the MiniMPL instrument. However, it should be possible to make the same observation with the other devices. It would be interesting to have the difference between day and night for the other instruments as well.

13. P.21 L.11-14: see comment above

14. Fig. 9, 10 and 11: Check caption and labels for consistent use of the term attenuated backscatter (comment P.16 L.11)

15. Supplement: Some links to the plots in the tutorial seem not to be working. In the online version it is correct.

**Technical corrections**

1. P.2 L.14 and 19: ACL –> ALC

2. P.10 L.7: mostly likely –> most likely

3. P.10 L.14: no –> not

4. P.16 L.12: check unit of the threshold: $m^{-1}sr^{-1}$

5. P.18 L.15: show –> snow
* * *

---

## Author Comment (AC1) · 14 Oct 2020

**Authors' response to the referee comments on *"Ground-based lidar processing and simulator framework for comparing models and observations"**

P. Kuma, A. J. McDonald, O. Morgenstern, R. Querel, I. Silber, and C. J. Flynn

14 October 2020

Dear Editor and Anonymous Referees,

We would like to thank the anonymous referees for their comments. Please find our response to the comments below. The original referee comments are marked in **bold** above our response. The revised manuscript and a latexdiff document highlighting the changes are appended at the end. The page and line numbers (P*xx*L*xx*) in our response refer to this most recent latexdiff document. References to sections, figures and tables in our response refer to the revised manuscript.

**Anonymous Referee #1**

**The paper fits to the scope of GMD however, before publication a few clarifications are required, and the structure must be revised (order of figures should be 1,2,3,..., outline of the paper), it seems that the reorganisation has been done very fast after the quick review .**

**General comments:**

**1. The requirements on simulating and/or evaluating ALC-signals depend on the application: for the determination of cloud bottom heights (CBHs) they are certainly quite different compared to particle backscatter coefficients. It should be made very clear in the paper, which application is aimed at and how the requirements for this application are fulfilled. An outlook on planned extensions can be given for some relevant applications not yet implemented.**

We want to clarify that we do not evaluate cloud base height in our manuscript, but rather the full cloud mask as determined by the cloud detection algorithm based on the attenuated volume backscattering coefficient. This approach provides a more comprehensive comparison between observations and models. It is more dependent on accurately calibrated absolute attenuated volume backscattering coefficient. The ALCF output also contains cloud base height, but it is simply determined from the lowest level of the cloud mask.

Aerosols are not currently implemented in the lidar simulator (this is also true for the original spaceborne lidar simulator) and also not implemented in the detection algorithm. We added a sentence to the abstract "While the correct focus of the framework is on clouds, support for aerosol in the lidar simulator is planned in the future." (P1L20) We also added text to Sect. 4 stating explicitly that aerosols are not implemented in the simulator and the focus is on cloud evaluation (P9L17) and text in Discussion and conclusions (P36L11).

**2. Section 2: The lidar simulator should be explained in more detail and whether the focus of ALCF1.0 is on clouds or aerosols or both. In the present state it seems to be clouds , as this is much easier to be treated: The numerical models/reanalyses provide the necessary information and the variability of the lidar ratios is comparable small. Low clouds typically consists of droplets so the consideration of non-spherical ice crystals is not highly important (provided that the focus is on CBHs only). Anyway: it is strongly recommended to include to consideration of optical properties of ice crystals into ALCF2.0 (phase functions are available since decades).**

We recognise that implementation of aerosols would be a desirable feature of the simulator. This is, however, a relatively complicated problem due to the relatively large variety of aerosol types implemented in different models

and would deserve its own paper. Also reanalyses generally do not provide aerosol concentrations in their standard output. The current focus of the simulator is on clouds. Aerosols affect the analysis of backscattering from clouds by attenuating the lidar signal before reaching the cloud. We do not expect this attenuation to significantly affect cloud detection unless the measurements are done in a location with high levels of aerosol concentration (P36L11).

As we note above, the current focus is on clouds and we made clarifications in the text (P1L20, P9L17). In the revised version we implemented backscattering from ice (Sect. 4.3, P9L20, P35L7), where the lidar ratio varies much more considerably than in liquid clouds. As we note above, cloud base height is not evaluated in the manuscript but rather the cloud mask.

The knowledge of the phase function of particular ice crystals is not enough for implementation of ice in the simulator because of the unknown distribution of shapes and size of ice crystals. Therefore, we decided to base the ice lidar ratio on global observations from CALIPSO (Garnier et al., 2015).

**The treatment of aerosols is not sufficiently explained: the backscatter signal depends on the aerosol distribution and their optical properties. Where is this input coming from in cases when the model/reanalysis does not provide aerosol information (NWP not necessarily consider aerosols). What about non-spherical particles – in the case of e.g. dust or volcanic ash the application of Mie theory is certainly not justified. Even if aerosol applications are not included in the paper this topic must be discussed (maybe as outline of the next version of ALCF). Chan et al. (2018) have demonstrated that there is an influence of particle shape on the backscatter profiles (and the intercomparison with model results), and that consideration of aerosols and their non-sphericity is possible.**

**Chan et al: Evaluation of ECMWF-IFS (version 41R1) operational model forecasts of aerosol transport by using ceilometer network measurements, Geosci. Model Dev., 11, 3807–3831, https://doi.org/10.5194/gmd-11-3807-2018, 2018.**

As we note above, aerosol is not currently included in the simulator, even though some models such as the UM potentially provide information about aerosol concentration. Our case studies are affected by relatively little aerosol. The atmosphere of New Zealand has little anthropogenic aerosol except for major cities especially in winter. Attenuation by aerosol is unlikely to be a significant factor for studying backscattering from clouds, which generally produce about an order of magnitude or more backscattering.

Aerosol affects the attenuated volume backscattering coefficient by producing additional backscattering on top of the molecular and cloud backscattering, as well by attenuating the signal. This could be partially avoided by filtering out time periods with high concentrations of aerosol identified manually (automatic identification would probably require some kind of a machine learning algorithm). In our datasets aerosol with volume backscattering coefficient of up to about $2 \times 10^{-6} \mathrm{m}^{-1} \mathrm{sr}^{-1}$ was present, spanning range of up to about 4 km. If we assume lidar ratio of 50 sr, this would produce two-way attenuation of up to about 50%. Therefore, this can be expected as the worst-case limit for accuracy of the molecular backscattering calibration in our dataset. For the purpose of determination of the cloud mask this should be still acceptable, as the cloud volume backscattering coefficient varies over multiple orders of magnitude (from approximately 2 to $200 \times 10^{-6} \mathrm{m}^{-1} \mathrm{sr}^{-1}$ in our dataset). We clarified this in Discussion and conclusions (P36L11).

The optical properties listed in Chan et al. (2018) would be a good start for implementation of aerosols in the simulator. However, more aerosols type might need to be calculated for models such as the UM, which provides a much larger variety of aerosol species in its output.

**3. Section 4: In the introduction of this section again the description of the treatment of aerosols has been forgotten.**

We we note above, we clarified this in the abstract and in the introduction of Sect. 4 (P1L20, P9L17).

**In section 4.1 (starting with the paragraph above Eq. 2) it should clearly stated whether the authors talk of clouds or aerosols. Aerosol size distributions typically are not described by gamma-distributions, and the**

application of Mie theory is often not adequate.

We replaced "atmospheric constituents" with "cloud droplets" (P11L16).

**Moreover, the authors should comment how they deal with the lidar ratio: when CBHs shall be derived the magnitude of the lidar ratio is not very important: the backscatter coefficient is such large that the uncertainty of the lidar ratio is more or less a second order effect. They should also comment on the consequences of the wrong description of ice crystals by the Mie theory: is this tolerable in view of the overall relevance of the lidar ratio in case of CBHs?**

As we note above, cloud base height is not evaluated in the manuscript but rather the cloud mask. The variability of lidar ratio of cloud droplets is relatively small (ranging between approximately 14 and 19 sr for effective radius between 10 and 30 $\mu$m and lidar wavelength between 532 and 1064 nm (Fig. 2b), i.e. variability of about 30%, this is potentially significant and we think it is justified to parametrise lidar ratio as a function of the effective radius and lidar wavelength in the simulator. Ice crystals have a greater variability of lidar ratio between approximately 20 and 50 sr (Fig. 2c), and therefore correct parametrisation of lidar ratio is even more important.

**Are the derived microphysics of the cloud particles (mainly size distribution) checked against the input used in the models/reanalyses. If it is available it should be consistent (ALCF vs. model).**

We showed that the size distribution assumption does not produce a significant difference in the lidar ratio (Fig. 2b). Therefore, we do not expect that it would produce a significant effect on the attenuated volume backscattering coefficient. In the future the size distribution assumption could be made consistent with each type of model and reanalysis (if this information is available). This kind of information is not generally present in model and reanalysis output.

**In Section 4.3 multiple scattering is briefly described. Of course it belongs to a full description of radiative transfer but in case of CBH-determination I don't believe that this is really relevant (the photons stay close to the optical axis). A clarifying sentence might be added.**

As we note above, cloud base height is not evaluated in the manuscript but rather the cloud mask. Previous studies identified potentially large multiple scattering correction required for a ceilometer with a similar field of view as our instruments (110–830 $\mu$rad) (O'Connor et al., 2004).

**4. Section 7: This section is quite long and should be separated into subsections. If done so 7.1 would deal with Fig. 4, Section 7.2 with Figs. 5-7, and Section 7.3 with Fig. 8 and so on.**

We split Section 7 into subsections "Cass", "Lauder", "Christchurch", "Backscattering on daily scales", "Cloud fraction and cloud albedo", "Molecular backscattering, aerosol backscattering and noise".

**In this context I suggest to delete Section 7.3 (Fig. 8): The arguments of the authors with respect to the cloud albedo are very difficult to understand and thus not convincing, and for me highly speculative: How is the effect of overlapping clouds in different levels considered?**

The effect of overlapping clouds in treated implicitly by the lidar simulator. It simulates cloud obscuration as would happen in reality for the given model cloud mixing ratios and cloud fraction. In that sense, the comparison of LR between observations and models is unbiased, but only if the effect of aerosol can be neglected. We agree that the lack of aerosol in the lidar simulator could cause a systematic bias in the results presented in Fig. 8. Therefore, we decided to remove Fig. 8 and the related discussion (Sect. 7.3 and P1L13).

**How accurate is the estimation of the lidar ratio (ice crystals)?**

The revised version of the simulator has parametrisation of lidar ratio and multiple scattering coefficient of ice crystals based on temperature (Sect. 4.3). Uncertainty still remains due to the fact that we used global empirical distributions derived from CALIPSO and there are more factors than temperature which affect the optical properties of ice crystals.

The calculation of the lidar ratio from the vertically integrated backscatter is not explained (in contrast to the extended description of r eff etc. in Section 4.1). How is obscuration of high clouds by low clouds considered, and so on. An ALC can provide a lot of useful information so it is not necessary to invent retrievals of further parameters associated with low confidence.

We added Sect. 4.1 which reviews lidar ratio and its derivation from the vertically integrated attenuated volume backscattering coefficient. This is an established parameter used by previous studies (O'Connor et al., 2004).

In connection with Figs. 5–7 it should be outlined why the backscatter coefficient is plotted and not the cloud mask. In the latter case the model will show clouds that might be obscured by low clouds in case of ALC-measurements. This could give further insight into the fundamental differences between remote sensing and modeling (and the problems associated with the resolution).

In the original Fig. 4–6 the lower limit for the attenuated volume backscattering coefficient was chosen the same as the threshold for cloud detection. Therefore, showing the cloud mask would not provide additional information. In the revised figures the lower limit is $0.1 \times 10^{-6} \mathrm{m}^{-1} \mathrm{sr}^{-1}$ and the cloud mask is plotted as dashed outlines. The input fields of cloud liquid water and ice mixing ratios and cloud fraction are in panels below. It should be clear how the obscuration by low cloud affects higher level clouds in the simulated attenuated volume backscattering coefficient.

In connection with Figs. 9–10 again the aerosol related questions from above show up.

Aerosol affects these figures by producing additional backscattering on top of the molecular and cloud backscattering, as well by attenuating the signal. This could be partially avoided by filtering out time periods with high concentrations of aerosol identified manually (automatic identification would probably require some kind of a machine learning algorithm). As we note above, in our dataset aerosol could produce additional attenuation of up to about 50% and this can be assumed to be the worst-case limit of the molecular backscattering calibration method, which is still good enough for absolute calibration for the purpose of cloud detection.

Specific/minor comments:

1. 2/8: aerosol optical depth should be replaced by particle backscatter coefficient: AOD is not a primary output of ALCs as it is an integral quantity, and depends on the lidar ratio.

We replaced this with "particle volume backscattering coefficient" (P2L6).

2. 2/14: thousands is a little bit exaggerated: each network rather consists of a few hundreds at maximum.

We replaced "thousands" with "hundreds" (P2L15).

3. 2/18: model evaluation : here a few references would be fine.

We added references to Hogan et al. (2001); Illingworth et al. (2007); Morcrette et al. (2012); Warren et al. (2018); Lamer et al. (2018) and Hansen et al. (2018) (P2L18).

4. 4/13 ff: check order!

We reformulated this paragraph (P4L17).

5. 5/18 (and throughout the paper): please avoid terms like backscatter : either backscatter coefficient, particle backscatter coefficient, attenuated backscatter or any other clearly defined physical quantity.

Where appropriate, we replaced occurrences of "backscatter" with "attenuated volume backscattering coefficient".

6. 6/1: in the near IR spectrum can be deleted, the same is true for in the visible spectrum in line 17.

We put "near IR" and "green colour in the visible spectrum" in parentheses (P7L1, P7L7, P7L18). We think this is still useful information, even though most readers might be well aware of this.

7. **6/2: the native resolution is indeed 5 m, but the typical output-resolution is 15 m.**

We clarified this with "vertical sampling resolution 5 m in the first 150 m and 15 m above" (P7L2).

8. **6/3: uncalibrated : this is a little bit misleading as the signals (to my knowledge) are calibrated against a standard instrument. This has been done to make all instruments within a network comparable. I don't know if this applies to the ceilometers in New Zealand as well.**

According to information in (Hopkin et al., 2019, Fig. 8, Fig. 13) the instruments are not all absolutely calibrated to the same reference. The instruments in our study are off-the-shelf instruments. Their calibration was done by the manufacturer. Therefore, determination of the calibration coefficient is necessary to get the absolutely calibrated attenuated volume backscattering coefficient. Without the calibration step the per-instrument default values of the calibration coefficient are used.

9. **6/18: is up to 5 m : what does this mean? Is the resolution coarser (10 m or so) or finer (3 m)?**

We replaced this with "The vertical resolution is 5–75 m and sampling rate 1 s." (P7L21).

10. **6/19: The instrument can be housed... : Why is this mentioned? It is true for the other instruments as well.**

This is mentioned because there is an option to operate this instrument without an enclosure, which is not an option with the CL31, CL51 and CHM 15k, which are permanently integrated with their enclosure. Another reason is that the enclosure has a scanning head (pointing vertically in our case studies). We clarified this in the text (P7L23, P24L3, P24L12).

11. **7/2 (and later): when giving the number of vertical levels it should be added how many levels are within the range of the ALCs, e.g. CL31. And the typical vertical resolution in this range should be mentioned.**

The number of model levels in the instrument range is generally dependent on the geographical location due to orography. We added a table listing the number of levels for each of the three sites and instruments (Table 7) and a reference to this table in the text (P22L12, P21L16).

12. **7/10: What does  single level  mean?**

We replaced this with "surface level" (P8L15).

13. **8/7: What is meant be  horizontally homogeneous ? The lidar equation is one-dimensional (along the pointing of the laser beam, typically vertically).**

We removed this part of the sentence because it is not strictly necessary. Horizontally homogeneous atmosphere assumption is commonly used in radiation transfer schemes and it simply means the atmospheric variables are a function of height only and horizontal dependence is neglected. This could be applicable with a lidar simulator if pointing at an off-zenith angle.

14. **8/21: usually lower : I would rather write  much smaller . The backscatter coefficient of (low) clouds is several orders of magnitude larger than Rayleigh scattering. The lidar ratio of Rayleigh-scattering should be mentioned.**

In our experience this is not the case. The molecular volume backscattering coefficient is about $1.54 \times 10^{-6} \mathrm{m}^{-1}\mathrm{sr}^{-1}$ near the surface at 532 nm (Fig. 8j). Clouds in our dataset produce backscattering from about $2 \times 10^{-6}\mathrm{m}^{-1}\mathrm{sr}^{-1}$.

15. **12/31 ff: I don't know if it is necessary to consider calibration constant from  past studies . Most ceilometers show calibration constants that are not really constant (temperature-dependent), and maybe some instruments have been set to a higher/lower sensitivity on purpose. If ALCF1.0 offers the option to calibrate signals (Rayleigh or cloud method) it is not required to rely on previous measurements anyway.**

We include these values as default both as a first approximation and for the convenience of users who do not require a high accuracy of calibration (e.g. if they do not want to compare observations with a model). The default

values are probably still relatively good as a first approximation, Fig. 8 and 13 in Hopkin et al. (2019) show that most instruments have a calibration coefficient within an order of magnitude or less.

**16. 14/24: It is certainly a good idea to define a absolute threshold for the cloud detection. However, $2 \times 10^{-6}$ m $^{-1}$ sr $^{-1}$ corresponds to an extinction coefficient of 0.04 km $^{-1}$ (assuming a lidar ratio = 20 sr), I would have expected a larger value. Maybe a few additional comments on the magnitude can be added (typical values for low water clouds).**

$2 \times 10^{-6} \mathrm{m}^{-1}\mathrm{sr}^{-1}$ was a value which was the best compromise for false positives and misses in our datasets, which were affected by relatively little aerosol. Users are advised to tune the threshold for their situation. In our datasets some clouds produced backscattering below $2 \times 10^{-6} \mathrm{m}^{-1}\mathrm{sr}^{-1}$. We added text explaining this point (P21L6). Temporal and vertical subsampling (5-min and 50 m, respectively, by default in the ALCF), removal of molecular backscattering and 5 standard deviations of noise before cloud detection reduce the number of false detections even at a relatively low detection threshold.

**17. 14/31: simulated backscatter too crudely : see comment 7/2.**

We added reference to the new table (P21L16).

**18. 15/25: Only the altitude of Cass is given here. To be consistent the altitude of the other stations should be given as well in this section.**

We added altitude to the description of the two other stations (P24L1, P24L5).

**19. 16/12: Typo in unit.**

We changed the units to $\mathrm{m}^{-1}\mathrm{sr}^{-1}$ (P24L20).

**20. 18/3 ff: When discussing Figs. 5–7, the panels of the lidar ratio are ignored. They are also not explained in the figure captions. What is the meaning of these panels? Can they be omitted?**

We replaced the lidar ratio panels with model cloud liquid water and ice mixing ratios and cloud fraction. We kept the lidar ratio panels under the observed attenuated volume backscattering coefficient (Fig. 4a, 5a, 6a). We also replaced it with effective lidar ratio. Previously true lidar ratio was calculated under the assumption of multiple scattering coefficient 0.7, but cannot be universally assumed for ice crystals, which are supported in the revised lidar simulator. In the revised manuscript, lidar ratio is discussed in Sect. 4.1. In addition, calibration using lidar ratio was already discussed in Sect. 5.2. We also added a paragraph at the end of Sect. 7.4 discussing the lidar ratio panels (P31L2). They are useful for determining if the attenuated volume backscattering coefficient is properly calibrated.

**21. 18/5: What is favourable ? Is it meant that individual intercomparisons of CBH between ALC and models are more or less impossible due to the different spatio-temporal resolution. So the focus must be on climatologies, and the provision of the true CBH and vertical cloud distribution for model-validation.**

As we note above, cloud base height is not evaluated in the manuscript but rather the cloud mask. By "favourable" we mean that even models with high spatiotemporal resolution (ERA5, UM) struggle to simulate clouds correctly on hourly time scales. Comparison of individual daily profiles is definitely possible and can uncover deficiencies in models (Kuma, 2020, Chapter 4). For models where only low resolution output is available (AMPS, JRA-55, MERRA-2), comparison of climatologies may be preferable. The individual daily profiles generated from instantaneous model output should still be comparable with observations if the limitations due to resolution are taken into consideration. The SCOPS subcolumn generator ensures that such a comparison is statistically unbiased.

**22. 18/27: Fig. 5l does not exist (typo). More examples like this are existing.**

We fixed the references to Fig. 5–7.

**23. 20/32: relatively poorly . Here it should be stressed that the sample is limited, thus general conclusions might be difficult.**

We added "in our time-limited local comparison" (P35L11).

24. 42: Figure caption should be revised: One description of day and night is sufficient.

We removed the definition.

**Anonymous Referee #2**

**General comments**

The paper deals primarily with comparing cloud distribution, the cloud base height and cloud layers from observations and models. However, the influence of aerosol on radiative transfer has to be considered in any case if quantitative statements are to be made. For the investigation of only the mentioned cloud parameters a calibration is not absolutely necessary except for the harmonisation of the data and to find a common cloud detection threshold. However, with a view to a next version of the framework, the already included parts such as multiple scattering, Mie and Rayleigh scattering calculations and the calibration itself provide already the basis to include the option of investigating aerosol optical properties as well.

The authors should emphasize that the focus in this version of the framework is on clouds and not on aerosol which has influence on the attenuated backscatter of observations and of simulated lidar signals.

As we note above in our response to Referee 1, we added clarifying text to the abstract and Sect. 4 (P1L20, P9L17).

Section 7: This is a rather long section and should be divided into subsections dealing with the different figures 4, 5-7, 8, 9-11.

As we note above in our response to Referee 1, we split this section into multiple subsections.

Fig. 5 it seems that for some reanalyses and models the cloud layer near the ground seem to fully attenuate the signal and no clouds above can be detected whereas higher clouds are visible in the observation. This is especially true for the models and reanalyses with very coarse resolution. However, the reanalyses and models may have calculated the clouds, but they are not visible in the simulated lidar signal. Have you compared the observed signal also with cloud masks or cloud fraction of the models? Is the picture different if you would plot other subcolumns than the first one?

As we note above in our response to Referee 1, the revised Fig. 5–7 contain panels with model cloud liquid water and ice mixing ratios and cloud fraction. If obscuration by low clouds happens in the model but not in observations, it means that the low clouds in the model are too opaque or cover too large area.

The lidar ratio subplots in Fig. 5-7 can be omitted since they don't appear neither in the text nor in the caption.

As we note above in our response to Referee 1, we replaced the lidar ratio panels with model cloud liquid water and ice mixing ratio and cloud fraction.

Fig. 8: Aerosol would have influence on the lidar ratio determined with the applied calibration method in this paper as stated on P.15 L15. Aerosol in the layers below the cloud would lead to an increase of the determined lidar ratio (decrease of LR $-1$) what can be seen in Fig. 8. Because aerosol is not considered in the used models here, they are not affected. Could you please comment on how you screened the data for aerosol and is its influence somehow considered in your calculations?

As we note in our response to Referee 1, we decided to remove Fig. 8 and the accompanying discussion because of the potentially systematic bias caused by the lack of aerosol in the lidar simulator.

**Specific comments**

1. P.2 L.4: ash is particulate matter as well. Thus, just aerosol is sufficient.

We removed this word (P2L2).

2. P.2 L.8: aerosol optical depth is an integrated quantitiy relying on the lidar ratio. ALCs are used to determine the particle backscatter coefficient: Change to particle backscatter coefficient

As noted in our response to Referee 1, we replaced this with "particle volume backscattering coefficient" (P2L6).

3. P.2 L.15: EARLINET is not a network of ALCs. Most of the instruments are not operated autonomously. The measurements must meet stringent quality control criteria and are performed for selected times. A better example would be PollyNET:
Baars et al.: An overview of the first decade of PollyNET: an emerging network of automated Raman-polarization lidars for continuous aerosol profiling, https://doi.org/10.5194/acp-16-5111-2016

We removed EARLINET and included PollyNET (P2L15).

4. P.6 L.19: instrument can be housed in a protective enclosure : This is not only true for the MiniMPL systems but also for the other ALCs to allow a 24/7 operation at any location.

As we note above in our response to Referee 1, we mention this because the enclosure is optional with the MiniMPL and contains a scanning head. This is clarified in the revised manuscript (P7L23, P24L3, P24L12).

5. P.8 L.10: As already mentioned, it should always be emphasized that the focus here is on clouds and that aerosol is not taken into account. This however would be necessary to perform the correct radiative transfer calculations in order to appropiately simulate the lidar signal.

As we note above in our response to Referee 1, aerosol is unlikely to affect cloud evaluation due to the relatively low attenuation caused by aerosol compared to the strong backscattering from clouds. Aerosol simulation is only possible with those models which provide aerosol concentration. We added text clarifying that aerosol is currently not implemented in the simulator (P1L20, P9L17).

6. P.13 L.1 and 2: Is the calibration constant a unitless quantity?

The units depend on the units of backscattering recorded by the instrument. The calibration coefficient translates between arbitrary units of the instrument and the units of the attenuated volume backscattering coefficient $(m^{-1}sr^{-1})$. We added an explanation (P18L20).

7. P.14 L.24: The cloud detection threshold of $2x10-6\,m-1\,sr-1$ corresponds to a particle extinction coefficient of $0.1\,km-1$ if assuming a lidar ratio of 50 sr which is not unusual for aerosol. This threshold seems to me rather low and could lead to misclassification of aerosol as clouds. This is even more true in the case of the MiniMPL system which is operated at 532 nm. Could you please comment on the observed magnitudes within liquid and cirrus clouds? Why is no adaption to the wavelength needed? Would it make sense to apply a height dependent threshold with lower values at higher altitudes to account for the lower attenuated backscatter of cirrus clouds?

As we note in our response to Referee 1, the threshold of of $2 \times 10^{-6} m^{-1}sr^{-1}$ was the best compromise between false positives and misses in our dataset. Users are advised to choose a threshold suitable for their situation if the aerosol contamination is higher for example. We clarified this in the text (P21L6).

In the revised version we implemented the option to couple observed backscattering with simulated backscattering and remove molecular backscattering from the observed attenuated volume backscattering coefficient before cloud detection is applied (P21L3, P24L24). This improves the results, especially with the MiniMPL which is affected by the largest amount of molecular backscattering at 532 nm.

Our dataset does not show a strong dependence of cloud attenuated volume backscattering coefficient on height to justify a height-dependent threshold, especially if molecular backscattering is removed prior to cloud detection.

8. P.16 L.11: Please check the use of the different terms attenuated backscatter , total attenuated backscatter , total volume backscatter coefficient and particle backscatter coefficient in the text, the figure captions

and labels. The measured range corrected lidar signal calibrated with the calibration constant is normally referred to as the attenuated backscatter.

As we note above in our response to Referee 1, we replaced the occurrences of "backscatter" with "attenuated volume backscattering coefficient".

**9. P.18 L.17, 19, 27 and 29: Please check figure labels for Fig. 5 in the text. There are no subfigures g, h, k, i, l.**

We fixed the references to Fig. 5–7.

**10. P.19 L.29: Again, as already mentioned, aerosol can not be neglected. Especially when using the molecular backscatter to determine the calibration coefficient, aerosol must be considered. As you can see in Fig. 9 and 10 the streak caused by molecular backscatter is too broad to retrieve an accurate calibration constant. This is mainly caused by aerosol and varying atmospheric conditions (temperature, pressure). I would omit here and in P.20 L.10 the proposal of a new method.**

We disagree that the molecular backscatter is too broad to retrieve an accurate calibration coefficient. For the purpose of cloud detection, a calibration coefficient accuracy better than a factor of 2 is still satisfactory, which is within the uncertainty range in Fig. 9 and 10 for at least the MiniMPL, where the peak is centred between 1 and $2 \times 10^{-6} \mathrm{m}^{-1} \mathrm{sr}^{-1}$ (Fig. 9e, i; Fig. 10e). In a more polluted atmospheric conditions it may not be possible to clearly identify the streak, but in general this method may still be useful in conjunction with the stratocumulus method, especially if a dataset does not contain suitable cases of stratocumulus clouds.

**11. P.20 L.9: Fig. 10 instead of Fig. 9**

We replaced the reference with the correct figure (Fig. 8 in the revised manuscript).

**12. P.20 L.21 and 24: You are only addressing the difference between day and night here for the MiniMPL instrument. However, it should be possible to make the same observation with the other devices. It would be interesting to have the difference between day and night for the other instruments as well.**

We added night and day for CL51, CL31 and CHM 15k. They differ relatively little due to low incoming solar radiation at near infrared wavelengths compared to 532 nm. We updated the text accordingly (P34L10, P35L27).

**13. P.21 L.11-14: see comment above**

We expanded the text to describe the nighttime and daytime results for CL51, CL31 and CHM 15k (P34L10).

**14. Fig. 9, 10 and 11: Check caption and labels for consistent use of the term attenuated backscatter (comment P.16 L.11)**

As we note above in our response to Referee 1, we replaced the occurrences of "backscatter" with "attenuated volume backscattering coefficient".

**15. Supplement: Some links to the plots in the tutorial seem not to be working. In the online version it is correct.**

We fixed this in the v1.0.0-beta.3 release.

[revised manuscript text omitted]